# Listening Beyond the Source: Exploring the Descriptive Language of Musical Sounds

**DOI:** 10.3390/bs15030396

**Published:** 2025-03-20

**Authors:** Isabel Pires

**Affiliations:** 1CESEM—Centre for the Study of the Sociology and Aesthetics of Music, School of Social and Human Sciences of the NOVA University of Lisbon, Nova University, 1069-061 Lisboa, Portugal; isabelpires@fcsh.unl.pt; 2IN2PAST—Associate Laboratory for Research and Innovation in Heritage, Arts, Sustainability, and Territory, 7000-809 Évora, Portugal; 3IN2PAST—Associate Laboratory for Research and Innovation in Heritage, Arts, Sustainability, and Territory, 1099-085 Lisboa, Portugal

**Keywords:** abstract sounds, intrinsic sound qualities, cross-modal linguistic associations, sound entity quality research

## Abstract

The spontaneous use of verbal expressions to articulate and describe abstract auditory phenomena in everyday interactions is an inherent aspect of human nature. This occurs without the structured conditions typically required in controlled laboratory environments, relying instead on intuitive and spontaneous modes of expression. This study explores the relationship between auditory perception and descriptive language for abstract sounds. These sounds, synthesized without identifiable sources or musical structures, allow listeners to engage with sound perception free from external references. The investigation of correlations between subjective descriptors (e.g., “rough”, “bright”) and physical sound attributes (e.g., spectral and dynamic properties) reveals significant cross-modal linguistic associations in auditory perception. An international survey with a diverse group of participants revealed that listeners often draw on other sensory domains to describe sounds, suggesting a robust cross-modal basis for auditory descriptors. Moreover, the findings indicate a correlation between subjective descriptors and objective sound wave properties, demonstrating the effectiveness of abstract sounds in guiding listeners’ attention to intrinsic qualities. These results could support the development of new paradigms in sound analysis and manipulation, with applications in artistic, educational, and analytical contexts. This multidisciplinary approach may provide the foundation for a perceptual framework for sound analysis, to be tested and refined through theoretical modelling and experimental validation.

## 1. Introduction

Auditory perception is a complex process involving physiological and cognitive mechanisms, as well as subjective interpretations. It plays a central role in shaping human interactions with the audible environment. While cognitive and physiological processes are fundamental to understanding how humans process sound, this study specifically examines how listeners empirically describe sounds using verbal expressions, focusing on semantic associations, and the intuitive ways in which people articulate auditory experiences.

A substantial body of research on auditory physiology and cognition, developed under controlled laboratory conditions, has provided valuable insights into the cognitive processes of auditory perception. However, these studies often focus on simple sounds in controlled environments, thus overlooking the complexities inherent in real-world listening. Most studies focus on identifiable natural or musical instrument sounds—where listeners can associate sounds with specific sources—thereby missing the distinct perceptual experiences elicited by abstract sounds that have no identifiable source.

In this article, abstract sounds are defined as auditory stimuli designed to exclude identifiable sources, causal indices, or contextual references. Unlike natural or musical instrument sounds, abstract sounds, free from external associations or familiar references, draw attention to the specificities of the sound wave. This allows the listener to engage deeply with the auditory experience and encourages exploratory behavior that may lead to deeper insights into sound perception.

By presenting these abstract sounds outside the context of music, the study ensures that participants’ responses reflect perceptual qualities rather than interpretative frameworks tied to music. As Schaeffer noted in 1966: “Having abandoned any reference either to instruments or to accepted values, all we have is collections of disparate sound objects. All we can do is compare them with each other, in all sorts of ways, in their contexture or their texture.” ([23]).

We explore how listeners rely on cross-modal linguistic associations by using a linguistic framework to describe abstract auditory experiences. Rather than focusing on multisensory perception in a literal sense, we examine how linguistic conventions rooted in other sensory modalities serve as intuitive tools for articulating auditory experiences. This approach bridges the gap between perceptual experience and verbal expression, highlighting the role of language in shaping our understanding of sound.

The abstract sounds used are intended to engage the listener’s attention on the intrinsic qualities of the sound, emphasizing perception over source recognition. This approach facilitates a deeper understanding of how the auditory stimuli enhance our understanding of auditory perception and the spontaneous verbal descriptors used to refer to them.

Although sound is inherently perceived through auditory mechanisms, the verbal descriptors used to articulate auditory experiences are often drawn from other sensory domains, such as vision or touch. This reflects a linguistic and cognitive strategy rather than true multisensory perception. The use of cross-modal linguistic associations highlights the limitations of language in describing abstract auditory phenomena and provides a metaphorical framework to bridge the gap between auditory perception and verbal expression.

Despite the growing body of research on the intuitive language used to describe sounds, there remains a significant gap in our understanding of how descriptive language applies to abstract sounds in exploratory listening contexts. There is a lack of knowledge about how the physical properties of sound waves influence auditory sensations in non-referential and abstract sound environments. Our contribution aims to contribute to address this gap by investigating the correlations between subjective sound descriptors and specific physical properties of sound waves, thereby improving our understanding of how listeners process and describe abstract sounds.

To gain insight into the triadic relationship between auditory perception, the physical characteristics of sound waves, and the verbal language used to express these perceptions, an interdisciplinary approach is essential. The verbal description of auditory experiences relies on cross-modal sensory interaction, where perceptual sensations generated in one sense (e.g., hearing) are intuitively correlated with other senses (e.g., touch or vision). As Pierre Schaeffer stated: “When we speak of a rough or matt, velvety or limpid sound we are comparing sound to a stone, skin, velvet, or running water. Analogy with microstructures seems much more justified, although applied to perceptions that have no tactile or visual links. There must be a reason why these comparisons are made in everyday speech, spontaneously and convincingly: it is because here it is not the objects of sight or hearing themselves that count but the way they are ordered.” ([23]).

This study distinguishes between two specific aspects: cross-modal perception and cross-modal linguistic associations. Cross-modal perception refers to the integration of sensory inputs from different modalities during the perceptual process. It describes how auditory experiences are influenced by sensory inputs from other domains as the brain processes and integrates information from different sources. For example, visual attributes such as brightness or tactile sensations such as surface texture can influence an individual’s perception of sound intensity, pitch, or quality of sound. This does not mean that non-auditory senses are actively involved in sound perception, but rather suggests that the brain integrates information from different sensory modalities to form an overall perception of auditory stimuli.

This perceptual experience gives rise to cross-modal linguistic associations in which metaphorical connections between senses play a key role in articulating experiences. For instance, expressions like “rough”, “granular”, or “smooth” are typically associated with tactile sensations, while words such as “bright”, “clear”, or “sharp” are often used to describe visual phenomena. These linguistic associations illustrate how individuals navigate through sound tactile and visual perspectives, helping to facilitate the understanding of abstract auditory phenomena by drawing on familiar sensory experiences. While these descriptors do not directly represent the physical properties of sound, they serve as practical tools shaped by cognitive and cultural frameworks.

Although the way people understand sound is influenced by their cultural context and life experiences, most participants in our study come from Western cultural contexts, which may provide a stable framework for the verbal descriptors used. However, it is likely that these verbal descriptors reflect a certain degree of universality in that they correspond to intuitive, cross-modal linguistic associations. It is commonly assumed that a rough surface is not perceived as smooth, a bright color is not perceived as dark, and a shiny object is not perceived as dull.

The abstract sounds used in this study are synthesized sounds designed to avoid identifiable sources, causal indices, or contextual references. This approach draws listeners’ attention to the intrinsic auditory qualities, which depend on spectral and dynamic properties. While these sounds have potential musical applications, they are presented outside a musical framework to minimize interpretation bias and encourage responses based solely on perceived qualities.

The subsequent discussion will explore correlations between subjective sound descriptors and the specific physical properties of sound waves, with particular emphasis on spectral and dynamic aspects. The aim is to contribute to the development of a framework that links perceived auditory qualities with their acoustic structure, providing insights into the perceptual dimensions of sound. These correlations contribute to an evolving understanding of auditory experience as a cognitive and exploratory behavior involving both perception and attention, and lay the foundation for applications in sound analysis, composition, and auditory training.

The next section provides an overview of the literature relevant to this study to contextualize the investigation. This is followed by a definition of abstract sound qualities and a discussion of their attributes and fields. The results of a field study conducted between 2011 and 2012[note 1], designed to investigate auditory perception in natural, open-field (non-controlled) listening environments, are presented. While this study includes dimensions related to the empirical descriptors used by listeners to refer to three fields of sound qualities as proposed by [17] ([17]) (see below for explanation), the discussion will focus on the two specific qualities of the field of matter: texture and color. These aspects have been chosen because of their relevance in understanding the cross-modal behavior of auditory perception and their potential to address critical issues at the intersection of the physical characteristics of sound waves and their impact on auditory perception.

## 2. Sound Qualities Perception and Descriptive Language: A Short Overview

Despite the substantial growth in studies on auditory perception, there remains a gap in research on the descriptive language used for abstract, non-referential sounds. While our understanding of how people conceptualize, describe, and integrate sensory experiences has been enhanced by research on intuitive verbal descriptions of sounds and cross-modal interactions in disciplines such as cognitive linguistics and music semiotics, this review focuses specifically on the perceptual mechanisms underlying verbal descriptions. It considers studies related to cross-modal interference in auditory perception, intuitive verbal descriptions of both abstract and identifiable sounds, and the concept of perceptual sound qualities.

While correlating auditory perception is related to the tactile or visual senses, much of the research on cross-modal perception has focused on auditory or visual impairments ([6]), synesthetic phenomena ([5]), or the study of musical or instrumental source-identifiable sounds ([24]). While Lawrence Marks’ study ([9]) also focuses on synesthesia, he also discusses the cross-modal linguistic associations between specific sound qualities and visual associations, highlighting the potential for cross-modal integration in sensory experience. Specifically, Marks noted that: “When speech or music has that peculiar capacity to conjure up visual images and qualities, the lightness or brightness of the visual image relates directly to the auditory brightness of the galvanizing sound, the size of the image to the auditory volume of the sound.” ([9]). Siedenburg et al. explain that verbal expressions used to describe sounds are “often conceptualized and communicated through readily available sensory attributes from different modalities (e.g., bright, warm, sweet) but also through the use of onomatopoeic attributes (e.g., ringing, buzzing, shrill) or nonsensory attributes relating to abstract constructs (e.g., rich, complex, harsh)” ([24]).

[28] ([28]), provided a comprehensive analysis of the cognitive processes that shape auditory perception, emphasizing factors such as expectation, structural integration, and sequencing in listeners’ descriptions of musical sounds. His framework suggests that listeners develop expectations based on prior knowledge and perceptual experience, which influence their verbal descriptions. [18] ([18]) extended this approach by examining how listeners engage with music in a temporal context, arguing that verbal descriptions are shaped by the enactive and contextual nature of listening. Reybrouck’s subsequent works ([19], [20]) further elaborate on the dynamic interaction between perception and verbal expression. Although these studies focus on musical or natural sounds with identifiable sources, they provide valuable insights into how cognitive processes shape auditory perception into structured linguistic expressions.

[8] ([8]) support the hypothesis that cross-modal interactions are deeply embedded in sensory processing and conceptual thinking, rather than merely being a by-product of the memory. The frequent use of visual metaphors in auditory descriptions suggests that the brain processes sound within a visual framework. Furthermore, the findings of [1] ([1]) show how participants use descriptors from other sensory domains when describing sounds, revealing a consistent association between sound textures and tactile textures. Although the authors used concrete, natural, or human sounds, participants in their study focused on the characteristics of the sounds themselves rather than their sources, which is why their research is relevant. For example, mechanical sounds were associated with harder, rougher, and moderately slippery tactile textures, while water-related sounds were associated with softer, smoother, and more slippery textures. Similarly, [14] ([14]) investigated the cross-modal perceptual associations between sound frequency and tactile height, focusing on participants’ responses to pitch (high- vs. low-frequency sounds) and the relative elevation of tactile stimuli presented at different heights. Participants’ hands touched the stimuli, and it was through this interaction that the association between sound and tactile elevation was made. Also, [26] ([26]) conducted a review of studies on the cross-modal correspondence between different senses, highlighting robust findings on the correlation of high-pitched sounds with higher visual positions or tactile stimuli, and low-pitched sounds with lower positions. This suggests that the conceptualization of auditory and visual/tactile stimuli as either “high” or “low” may be a product of spatial metaphors.

[11] ([11]) made a significant contribution to sound quality research by establishing a correlation between acoustic properties and perceptual dimensions of timbre. Their findings also suggested that listeners use cross-modal linguistic associations, such as visual or tactile analogies, to describe auditory experiences. Other studies ([7]; [29]) have examined the role of intuitive verbal descriptions and vocal imitations in communicating sound qualities. These studies show that people often rely on these descriptors to convey the qualities of identifiable, everyday sounds, but that they are less effective for abstract, non-referential sounds. This limitation highlights the need to develop approaches that correlate auditory sensations with the intrinsic characteristics of sound waves, particularly in abstract sound perception.

[4] ([4]) examines how the brain organizes sensory information across modalities, emphasizing the similarities between visual and auditory perception, and underscoring the importance of cross-modal integration. He suggests that verbal descriptors from one sensory modality can influence the perception of another. For example, regarding timbre, Handel states: “The descriptors for timbre often are cross-modal. Timbre is described in visual terms: sounds with higher spectral centers are bright, clear, active, or colourful while sounds with lower spectral centers are dark, dull, wooden, or colourless. But timbre also is described in textural terms: rough or sharp versus smooth or delicate, warm versus cold, or heavy versus light.” ([4]).

Other researchers ([22]; [30]) have investigated the impact of linguistic descriptions on auditory perception, focusing on metaphorical terms such as “brightness” and “roughness”, and highlighting the semantic crosstalk in timbre perception ([30]; [31]). These findings suggest that overlapping associations make the articulation of abstract auditory qualities highly complex further, highlighting the need for further research in this area. The use of visual and tactile descriptors to convey auditory experiences is better understood as a metaphorical transduction, where listeners rely on learned associations across sensory modalities ([17]). Consequently, cross-modal interactions play a central role in shaping auditory perception. As demonstrated by [10] ([10]), visual stimuli have been shown to alter auditory discrimination, while [15] ([15]) have shown how visual cues influence auditory processing at both the perceptual and neural levels.

Systematic studies, such as those conducted on psychological measurement of sound evaluation ([27]), have contributed to the development of methods to capture listeners’ subjective perceptions. Other research ([13]) has focused on perceptual qualities such as “roughness” and “density” in complex auditory textures, advancing our understanding of semantic mappings. [12] ([12]) emphasized the value of abstract sounds in perceptual research and argued for methods that capture intrinsic auditory qualities. Canelon’s research ([2]), based on Schaeffer’s Typomorphology ([23]), examined also the classifications of abstract sounds used by sound designers and emphasized the need for a standardized vocabulary.

## 3. Sound Qualities in Perception: Cross-Modal Perception and Fields

Adapted from the concept of fields in physics, [17] ([17]) developed a framework that categorizes sound entities based on their perceived qualities. This framework is also based on Pierre Schaeffer’s Typomorphology ([23]) and Denis Smalley’s Spectromorphology ([25]), and it represents how sound qualities interact locally to contribute to an interrelated, multidimensional auditory experience. It provides a foundation for examining correlations between auditory sensations and sound wave characteristics.

In the domain of auditory perception, the fields are matter, form, and position, and they correspond to distinct perceptual domains, each of which encompasses specific attributes derived from the physical properties of sound waves. These fields organize metaphorical descriptors into categories that encompass different perceptual dimensions of sound, providing a structure for cross-modal perception and semantic analysis. The interaction of these fields shapes the listener’s auditory experience. Schaeffer’s empirical approach illustrates this process: “Imagine that we can ‘stop’ a sound to hear what it is like, at a given moment in our listening: what we can grasp now is what we will call its matter, complex, fixed in the tessitura and in subtle relationships with the sound context. Now listen to the history of the sound: we become aware of the evolution in duration of what had for an instant been fixed, a journey in time that shapes this matter.[note 2]” ([23]). This perspective emphasizes the ability to analyze sound both in a fixed state and as a dynamic temporal evolution, reinforcing the interplay between matter, form, and position in auditory perception.

The concept of perceptual “sound quality”[note 3] involves attributes that enable listeners to discriminate and interpret sounds beyond conventional acoustic measures. Cross-modal sensory fields, which link auditory, visual, and tactile modalities, reveal how listeners intuitively associate sound qualities with other sensory properties, thereby shaping the auditory experience. Studying these structured perceptions provides insight into how listeners describe sound and how these descriptions relate to the physical properties of sound waves.

### 3.1. Defining Perceptual Sound Quality

In auditory perception, “sound quality” refers to the intrinsic characteristics of sound that go beyond conventional acoustic measures such as pitch or loudness. This facilitates the identification and interpretation of sounds by the listener. [16] ([16]) theorized that “quality” should be regarded as a “pure potentiality” arising from the physical properties of an object or phenomenon. Applying Peirce’s ideas to the quality fields of sound entities, Pires states that “[q]uality, therefore, does not depend on the subject’s perception of it; quality exists in objects, it depends on their particularities. Quality does not need to be measurable, it only needs to be perceived. A quality is a way of being, it is what is perceived as characteristic of an object, a phenomenon or an entity[note 4].” ([17]). Thus, in auditory perception, “quality” embodies the intrinsic characteristics or attributes of a sound, such as “bright”, “dense”, “rough”, or “smooth”. Perceptual sound qualities depend on the structure of the sound wave and are associated with specific physical properties—such as frequency, spectral structure, intensity, spatial position, and their evolution over time—that contribute to the distinctive perceptual identity of a sound.

These qualities are autonomous in nature, meaning they do not transform into each other, but they can influence each other. For instance, the pitch of a sound remains distinct from its timbre, even though intensity or texture can influence pitch perception. This autonomy emphasizes that these qualities are inherent in the sound itself, and perception reveals them, providing a foundation for the emergence of cross-modal perceptual fields of sound quality.

### 3.2. Emergence of Cross-Modal Perceptive Fields in Sound Quality

The analysis of the verbal descriptors used by the listeners tends to confirm the organization of the perceived sound qualities into three primary fields: matter, form, and position. This tripartite framework provides a structured approach to understanding the multidimensional nature of auditory experience, highlighting how listeners describe and categorize sounds. These descriptors reflect linguistic patterns in auditory perception, rather than direct cross-modal processing.

This approach relies on metaphorical language, reflecting the natural tendency to describe auditory qualities using terms from other sensory modalities, a phenomenon widely documented in cross-modal perception ([8]; [26]). For example, terms such as “brightness”, “roughness”, or “transparency” serve as metaphors, transferring sensory concepts from other domains to describe auditory phenomena. These descriptors do not imply literal multisensory engagement but illustrate how familiar sensory experiences are used to articulate auditory experiences. This approach is consistent with Schaeffer’s listening criteria outlined in his Treatise on Musical Objects ([23]), which emphasized the autonomy of auditory perception by isolating sound objects from external causal associations. Rooted in the multidimensional perspectives of auditory perception proposed by [23] ([23]), [3] ([3]), and [25] ([25]), this framework provides a solid theoretical foundation for this study.

The use of cross-modal descriptors highlights the perceptual richness and multidimensionality of sound. While the descriptors provided to participants were based on the established literature, listeners were also encouraged to suggest additional terms, ensuring empirical flexibility. This approach provides a reliable framework for exploring auditory qualities and their resonance with broader sensory metaphors, thereby advancing both theoretical and empirical understanding of sound perception.

#### 3.2.1. The Field of Matter

The field of sound matter refers to the intrinsic physical and perceptual properties of sound, including texture, density, and color, which define its material quality. These qualities are related to the internal organization and evolution of the sound wave structure. Based on Schaeffer’s Typomorphology ([23]) and Smalley’s Spectromorphology ([25]), sound matter includes both measurable components, such as frequency, amplitude, and timbre, and perceptual dimensions such as textural depth, density, and timbral complexity.

For example, texture reflects the granularity or smoothness of a sound, density its perceived compactness, and color its spectral richness. The perceptual experience of sound is shaped by these interdependent qualities, which influence each other and form the basis for sensations of form and position. This approach, emphasized in Schaeffer’s concept of “reduced listening”, isolates the sound matter as an autonomous phenomenon, independent of its source or cause, and emphasizes its intrinsic characteristics.

Sound matter evolves over time through processes such as granular synthesis, demonstrating how texture and spectral composition can be continuously altered, closely following Smalley’s concept of “spectromorphological growth”. Our study focuses specifically on the qualities of color and texture, providing insights into both the perceptual and physical dimensions of sound matter.

#### 3.2.2. The Field of Forms

The field of sound forms is concerned with the dynamic shapes, contours, and morphological characteristics of sound that reflect its temporal evolution and structure. As Rey notes, “Form is used to describe the shape or appearance given to a material thing [...]. We are dealing here with exteriority, with a configuration that is visible in one way or another. It is above all the contours with which we are concerned—in a sense close to morphology.[note 5]” ([21]). This concept also resonates with Schaeffer’s notion of the “sound mass profile” ([23]) and Smalley’s Spectromorphology ([25]), both of which emphasize the evolving contours or patterns within a sound’s progression, shaped by articulation, internal coherence, and temporal change.

Sound forms are perceived as abstract qualities that arise from relationships within the structure of a sound over time. Unlike visual forms, which are static and spatial, sound forms are inherently temporal, manifesting dynamically and unfolding as the sound evolves. Key elements such as pitch, intensity and spectral energy shape the perception of sound forms, allowing the listener to discern patterns that may appear distinct, fluid, structured, or amorphous. The temporal dynamics of sound reveal its evolution, offering insights into its organization, morphology, and unfolding nature. Smalley’s discussions of spectral shapes further highlight the role of forms in shaping auditory experience, underscoring their dynamic and perceptual significance.

#### 3.2.3. The Field of Positions

The field of sound positions relates to the spatial and directional aspects of sound as perceived within an auditory environment. As Smalley points out, sounds “have directional tendencies which lead us to expect possible outcomes” ([25]), whether they are static, directional, or chaotic. This perception is shaped by the relationship between the sound source, the listener, and the intrinsic properties of the sound.

Spatial qualities such as position, distance, mobility, and directionality define the positional attributes of sound. These attributes are influenced by physical factors such as intensity, spectral filtering, and the balance between direct and reverberant sound. Environmental factors, such as the acoustics of the room, and artificial manipulations, such as stereo effects, can further shape the spatial perception of sound and enhance the listener’s sense of the auditory environment.

Positions concern the spatio-temporal placement of sound, with localization and trajectory being key elements. Smalley’s Spectromorphology provides a framework for understanding how sounds occupy and move through space, emphasizing their mobility and interaction within a perceived spatial field. While mathematical models can define positions in virtual or composed spaces, perceptual experiences are often influenced by individual auditory orientation and the acoustics of the environment. The multidimensional nature of positions introduces a degree of complexity to their perception, where spatial and temporal dimensions interact to shape the listener’s experience.

### 3.3. The Qualities and Attributes of Sound Matter Field

The field of sound matter encompasses both the physical dimension, which includes measurable properties of the sound wave such as spectral content, amplitude, and their temporal evolution, and the perceptual dimension, which examines how these physical characteristics are translated into sensory experiences. Sound matter encapsulates the intrinsic properties of sound as perceived by listeners, influenced by their sensory memories, which lead to cross-modal associations between sound characteristics and descriptive terms rooted in past sensory experiences.

The field of sound matter is defined by three qualities: color, texture, and density.

*Color* embodies the harmonic and spectral richness of sound and shapes its timbral identity. Attributes such as “bright” and “dull” are indicative of color.*Texture* encompasses sensations ranging from granular, where individual sound particles are discernible and evoke roughness, to smooth and continuous, where no elements disrupt the sense of uniformity.*Density* refers to the compactness of internal sound structures at a given moment, or how they evolve over time, influencing perceptions of fullness or sparseness relative to other sounds. For example, terms such as “opaque” and “transparent” are often associated with density.

These qualities are interdependent and influence each other in perceptual experiences. For example, the perceived brightness of an object’s color can influence the perception of its texture, and this principle also applies to sound, where the brightness of a sound’s color can influence the perception of its texture, demonstrating their interdependent nature.

As mentioned earlier, this study focuses on the qualities of texture and color within the field of sound, excluding density due to methodological constraints. Specifically, the study presented isolated sounds for auditory evaluation, making it difficult to assess density-related qualities such as transparency or opacity, which require the comparison of multiple sounds. By prioritizing texture and color, the study ensures methodological rigor and meaningful analysis, leaving density for future research in contexts that allow evaluation of its perceptual effects.

#### 3.3.1. Sound Color

Sound color refers to a perceptual quality of sound that is closely related to its spectral content. It encompasses the timbral characteristics that result from the distribution and interaction of partials within a sound wave, contributing to its distinct auditory identity. Analogous to color in visual stimuli, the auditory perception of sound color evokes experiences shaped by the harmonic and inharmonic relationships between the components of the sound wave. This perceptual quality arises from the inherent spectral composition of the sound, rather than from causal references or specific sources. The concept of timbre thus extends the concept of timbre by emphasizing the spectral attributes that define the qualitative nature of a sound, such as brightness or darkness. While timbre is often associated with the characteristics of its production source (natural or instrumental), sound color in the context of abstract sounds focuses on the internal organization of the sound wave, specifically its spectral structure and evolution.

Sound color can be mapped perceptually along several axes, with brightness being a primary dimension. This axis enables the differentiation of sounds based on their spectral composition, ranging from dark to bright. Sounds characterized by a prevalence of high-frequency harmonics are perceived as “bright”, while those with limited, inharmonic, and predominantly low-frequency content are described as “dull” or “dark”. The increase in brightness is associated with high-frequency components, while low frequencies contribute to a darker quality. This suggests that sound color reflects the intrinsic spectral structure of sound, perceived along a continuum from dark to light. Four key attributes of sound color have been identified: brightness, dullness, darkness, and clarity. These attributes provide a framework for describing the spectral qualities of sound derived from its internal content and organization.

A sound is perceived as “bright” when it has a strong emphasis on high-frequency spectral energy, often characterized by crispness and sharpness. The higher the frequency content, the brighter the sound, such as the ringing of a bell or the sound of a violin.A sound that is perceived as “dull” lacks high-frequency energy, giving it a muted or subdued quality. These sounds have a spectral content predominantly in the low to mid frequencies, with little or no high-frequency emphasis, and are perceived as dull. These sounds feel less sharp and more rounded, like the low hum of a foghorn.A sound perceived as “dark” has a spectrum dominated by low frequencies, making it feel heavy or grounded. A bass drum or the low growl of an instrument is perceived as dark because of the dominance of lower frequencies.“Clarity” refers to the perceived distinctiveness of a sound, often linked to its spectral composition. Clear sounds tend to have well-defined harmonic content with minimal overlap or masking by other frequencies. Sounds with good clarity allow the listener to easily distinguish between different components, such as a pure piano note or a clean vocal recording.

These attributes allow sound color to be mapped perceptually from dark to light, providing a comprehensive framework for describing the spectral qualities of a sound. These attributes reflect the interaction of high and low frequencies and the listener’s ability to perceive the distinctiveness of the internal structure of the sound.

#### 3.3.2. Sound Texture

Sound texture is defined as the auditory sensation produced by the organization and interaction of the internal components of a sound over time. This is manifested as perceived granularity, smoothness, or roughness. This results from the spectral interaction of its micro-components and is a dynamic concept influenced by factors such as the density, distribution, and articulation of the sound wave partials. To illustrate this point, consider the auditory experience of granular synthesis, which typically yields sounds characterized by a “rough” texture due to the clustering of grains. In contrast, the simplicity of a sine wave conveys a “smooth” texture. Texture, like color, captures the tactile or visual sensations generated by the perception of a sound, providing a vivid auditory impression of its internal structure.

In auditory perception, sound texture refers to both the surface and internal structure of sound, often evoking cross-modal linguistic associations with tactile or visual experiences. This concept reflects the materiality of sound, shaped by an exploratory approach to listening that brings the listener into contact with its structural nuances. The perceptual descriptors of sound texture, such as smooth or rough, capture the listener’s sensory experience as influenced by the internal composition of the sound.

We have identified five key attributes for the analysis of sound texture: smoothness, roughness, granularity, striation, and iterativeness. Each of these attributes provides a specific sensory reference that contributes to the listener’s perceptual representation of the material of the sound. They are explained as follows:A “smooth” sound is perceived as continuous and flowing, with no abrupt changes or harsh interruptions. The spectral content is stable or transitions gradually, providing a steady, uniform auditory experience. An example would be a sine wave, where the sound is even and free of jagged variations.A “rough” sound is characterized by irregular, erratic micro-fluctuations in the amplitude or frequency of each spectral component. The irregularity in its waveform results in a jagged, abrasive, or harsh auditory sensation due to these unpredictable variations in its spectral content.A “granular” sound is made up of small, discrete sound particles or “grains” that are superimposed or densely packed. These sounds have a textured, “grainy” quality, often resembling a series of tiny clicks, bursts, or noise-like sounds that, when combined, form a continuous but textured sound. Granular synthesis specifically creates this texture by arranging small grains of sound into a cohesive auditory experience.“Striated” sounds show a regular, repeating pattern or structure, similar to stripes or lines. These sounds often have a rhythmic, periodic quality, such as the hum of a machine or the sustained note of an instrument. The rapid repetition and cyclical nature of the internal elements of the sound create an organized, patterned texture over time.An “iterative” sound involves the repetition of a pattern or structure over time, often with slight variations, iterations. But unlike striated sounds, which exhibit regularity, iterative sounds tend to evolve slightly with each repetition. They can evoke a sense of rhythm or repetitive motion, but with subtle shifts or changes in each cycle.

These texture attributes, or perceptual descriptors, help listeners to categorize the auditory materiality of sounds and provide a framework for conveying the nuances of sound texture.

## 4. The Sound Entity Quality Research Survey (SEQR)

The results presented in this section are derived from the Sound Entity Quality Research Survey (SEQR), conducted in 2011–2012 as part of my postdoctoral research on the auditory perception of complex, synthesized sounds in electroacoustic music.

The auditory stimuli were deliberately crafted to eliminate identifiable sources, causal indices, or contextual references. Tools such as CSound were used to create abstract sounds with intricate structures, focusing on their inherent qualities, including spectral, textural, and dynamic properties. This approach ensured that participants’ attention was focused on the characteristics of the sounds, rather than associating them with musical or environmental related sounds.

The SEQR was designed to investigate auditory perception in natural, open-field settings, as opposed to controlled laboratory environments. This allowed participants to interact with the sounds and respond to the survey without the need for specific instructions regarding listening conditions. The survey was conducted internationally, targeting academic communities with an interest in music, while remaining open to all participants. The survey was administered online via the platform wwwblu-world.net/[note 6], allowing for global participation. It consisted of two main sections: (1) demographic and professional characterization of the participants; and (2) assessment of auditory perception using predefined descriptors, with the possibility for participants to suggest additional descriptors. The survey concluded with questions designed to assess the participants’ opinions on the relevance of the study.

The two parts of the survey were:Participant Characterization and Survey AssessmentListening and Responding.

A total of 42 sounds were grouped into the three categories: matter, form, and position. The selection criteria prioritized a variety of spectral and dynamic properties to ensure a comprehensive exploration of perceptual qualities. The aim was to assess how participants use cross-modal descriptors to qualify abstract sounds and to explore correlations between subjective sound descriptors (e.g., “raw”, “bright”), and how these can be correlated with objective physical attributes such as spectral and dynamic properties.

### 4.1. Participant Characterisation

The survey collected demographic data, including gender, age, nationality, education, and professional background, with a particular focus on participants’ experience with music and sound processing software. The responses showed a diverse group of participants in terms of age, nationality, education, and profession, with a notable representation from music-related and sound processing fields. The data collected were analyzed to understand its potential impact on perceptual responses[note 7].

Participants were split into the following categories:Amateurs: Non-professional musicians who could indicate their preferred musical genres (e.g., classical, electroacoustic, pop, jazz).Professionals: Music or sound professionals, who could specify their role (e.g., composer, performer) and primary musical genres.

Participants were informed that the survey adhered to ethical guidelines for data protection, and that no personally identifiable information would be collected, thus ensuring confidentiality. By completing the survey, participants accepted the privacy conditions. A contact email address was provided to ensure transparency and to answer to any questions.

Incomplete responses were excluded to ensure data integrity. After careful evaluation, 87 surveys were initially validated based on completeness. To avoid bias in the subsequent analysis, surveys with inconsistencies in critical sections were also excluded from the final dataset. The final analysis was based on 70 participants who fully completed the sections related to sound quality analysis, ensuring meaningful and consistent data. The demographic data presented below includes all 87 surveys and provides a general overview of the participant pool, while the sound perception analysis is based on the 70 validated responses.

The demographic characterization showed that 68% of the participants identified as male and 32% as female, 60% were professional musicians, and 40% were amateurs. The age distribution is shown in Figure 1, with 52% of the participants in the 18–35 age range.

The geographical distribution of participants shows that they come predominantly from Portugal (48.3% by nationality; 52% by place of residence) and France (27.6%), reflecting the original distribution network of the survey (Figure 2).

### 4.2. Survey Assessment

The final questions, completed after the Listening and Responding section, assessed participants’ awareness of sound qualities, their potential applications in sound or music studies, and the relevance of the research.

The questions were as follows:Q1: Do you pay attention to sound qualities other than identifying the sound source? [Yes/No]Q2: Do you use music software? [Yes/No]Q3: Do you use sound manipulation software? [Yes/No]Q4: Would a perceptual sound manipulation interface be useful? [Yes/No]Q5: Could perceptual sound studies have applications beyond music and sound manipulation? [Open-ended question]

Figure 3 shows that the responses indicate a high level of interest in this area of research, with approximately 90% of participants responding in the affirmative to both Q1 and Q4. Analysis of these responses suggests potential for future applications, particularly in the development of user-friendly perceptual interfaces.

Regarding the open-ended question (Q5), 24 out of 87 responses were considered valid, yielding a response rate of 27.6%. Of these, only one participant considered that such studies have no application outside music, while six participants simply answered “Yes”. The analysis of the remaining responses identified five key areas where perceptual sound studies could have applications beyond music:Artistic and Creative Applications: Enhancing sound production for artists, including creators and performers, sound design for film, ambient sound creation, and optimizing soundtrack production tools for both amateurs and professionals.Educational and Therapeutic Uses: Applications in music education and therapy, in particular to improve auditory perception and sound quality recognition.Technological Advancements: Development of user-friendly sound manipulation interfaces for marketing, human–computer interaction, and ecological sound design.Scientific and Analytical Applications: Advancing data sonification, the semiology of music, and the study of sound–image relationships.Human Perception Insights: Investigating the brain’s perception of sound and improving the link between composers’ intentions and audiences’ perceptions.

This characterization enriches the data analysis by providing valuable context for interpreting participants’ responses. While the survey engaged a diverse group, the relatively small number of validated surveys (70) introduces some limitations, highlighting the need for future studies with larger, more representative samples. Nevertheless, given the scarcity of research on the perception of complex, abstract sound stimuli, the results of this study remain significant for advancing both theoretical and practical approaches to sound analysis and electroacoustic research.

### 4.3. Listening and Responding

The 42 abstract sounds used in the survey, each lasting between 2 and 10 s, were crafted to eliminate causal indices, and were organized into three distinct categories: matter, form, and position. The objective was to explore how listeners perceive and describe different aspects of abstract sound.

Matter Field: 18 sounds assessed based on internal or surface qualities, including a 1000 Hz simple sine wave as a test sound.Form Field: 12 sounds evaluated for shape or contour characteristics.Position Field: 12 sounds rated for spatial qualities, such as perceived position and movement.

Participants selected descriptors from pre-defined lists tailored to each category and could suggest additional descriptors. This design provided insights into how participants intuitively link auditory sensations with cross-modal perceptual qualities (e.g., tactile or visual terms) when describing abstract sounds.

## 5. Intrinsic Sound Qualities: SEQR Survey Results Analysis

As mentioned above, this analysis focuses on the area of matter, specifically on two of its qualities: color and texture. The results of the SEQR survey on the auditory perception of abstract sounds are presented here, with an emphasis on the correlation between subjective descriptors and objective sound properties. The analysis highlights cross-modal perception and shows how participants used cross-modal linguistic associations (e.g., “rough”, “bright”) to describe intrinsic sound qualities.

To aid interpretation, brief descriptions of each sound profile are given in Table 1. The survey results are presented in tables by sound quality: color (Table 2) and texture (Table 3). Each table shows the percentage of responses for each attribute associated with each sound, allowing the distribution of descriptors chosen by participants to be examined. This analysis reveals key patterns and correlations between descriptors and the physical characteristics of each sound, followed by a more detailed examination of selected pairs. These pairs were chosen based on cases where the perceptual results appeared to be contradictory, for example, where two similar sounds were described by very different descriptors, or where very different sounds were described by similar descriptors. The analysis is performed separately for color and texture.

Given the complexity of analyzing all possible combinations of descriptors for all sounds in the field of sound matter, this article focuses on specific examples that illustrate different case studies. The pairs discussed in Section 5.1 (color analysis) and Section 5.2 (texture analysis) were selected based on several criteria: pairs of similar sounds with clear perceptual descriptors, pairs of very different sounds with similar perceptual results, pairs of sounds perceived as equally ambiguous, and different sounds with different descriptor associations. A similar approach was used in Section 5.3, where the combined perceptual results of color and texture were analyzed.

**Table 1 behavsci-15-00396-t001:** Descriptions and generation methods for the 18 sound samples analyzed in the matter field of the SEQR survey, including a short profile of each sound[note 8].

Sound	Profile	Generation Method
**#1**	Sine wave. (1000 Hz)	Simple generator; CSound
**#2**	Near-harmonic spectrum with two prominent spectral peaks in the medium register	Modulated additive synthesis; CSound
**#3**	White noise, slightly filtered in a medium-low register	Noise generator; CSound
**#4**	Natural noise recording, processed to remove causal references	Natural noise manipulated; Audacity/CSound
**#5**	Granular texture generated from a recorded natural sound; medium-low register	Granular synthesis; CSound
**#6**	Buzzing sound in a medium register	Additive-based hybrid synthesis; CSound
**#7**	Similar to Sound #6, in a medium-low frequency range	Additive-based hybrid synthesis; CSound
**#8**	Equivalent to Sound #7, with slight amplitude modulation	Additive-based hybrid synthesis; CSound
**#9**	Medium-register sound with rapid pulsation, evoking a flutter	Additive-based hybrid synthesis; CSound
**#10**	Grainy, rustling sound akin to small wood shims, medium-high register	Granular synthesis; CSound
**#11**	Near-harmonic spectrum in a low register	Additive synthesis; CSound
**#12**	Buzzing sound in the medium-high register, with a fluttering, striated texture	Granular-based synthesis; CSound
**#13**	Slow, deep tremolo-like oscillation in a near-harmonic spectrum, medium register	Additive-based hybrid synthesis; CSound
**#14**	Buzzing sound in the medium-high register	Heavily processed natural sound; Audacity/CSound
**#15**	Liquid, granular texture reminiscent of large water droplets on wood, medium-low register	Processed natural sound with granular synthesis; CSound
**#16**	Granular sound with tiny grains in a thin medium register band, creating a pitch sensation	Granular synthesis; CSound
**#17**	Iterative texture in a thin medium register band	Granular synthesis; CSound
**#18**	Similar to Sound #16, with a broader medium register band evoking a white noise sensation, with no discernible pitch	Granular synthesis; CSound

The table above presents descriptions and generation methods for each sound sample analyzed within the matter field of the SEQR survey. It highlights distinct characteristics such as texture, register, and synthesis technique for each sound profile.

The discussion that follows examines the relationship between color and texture by exploring the relationship between sound wave characteristics and perceptual attributes from the survey in a selection of sound pairs. By calculating the Euclidean distance between these pairs, we integrate texture and color into a unified analysis. We will explore cases where sounds with distinct color attributes appear similar due to shared texture qualities, and vice versa. It is important to note that the “Other” option from the survey was excluded from the Euclidean distance calculations, as it was deemed insufficiently meaningful and potentially distorting the analysis of other attributes.

Euclidean distances were calculated by determining the squared differences between each pair of attribute values (e.g., color and texture descriptors such as “bright” or “rough”) and then taking the square root of the sum of these squared differences. This method provides a quantitative measure of the similarity or dissimilarity between the sounds, expressed as a percentage distance. As no normalization was applied, the distances are directly influenced by the input values, which represent the perceived incidence of certain qualities. The resulting values reflect the relative perceptual distances between the sound samples, measured in percentage points. The Euclidean distance serves as a simplified metric for quantifying perceptual differences based on the linear difference between descriptors, allowing for a straightforward measure of similarity or contrast between pairs of sounds, expressed in percentage-based distances, without the need for advanced statistical models.

The selected sound analyses are presented as images composed of four layers, each representing different physical characteristics of the sound, allowing detailed visual comparison without the need for audio playback. All analyses were performed in Sonic Visualizer, using the Vamp SDK Example Plugins, in particular the Linear Frequency Centroid plugin. To ensure consistency, all sound analyses were parametrized to ensure that the comparison remains unaltered and the visualizations remain reliable, including FFT window sizes and visualization scales that were carefully selected and maintained across all examples.

The four layers of the sound analysis displays are as follows:Waveform and Linear Frequency Centroid: The first layer shows the sound wave (in blue) and the linear frequency centroid (in green), providing a direct view of the amplitude and pitch variations over time. This layer has been analyzed using the Vamp Spectral Centroid plugin on a linear frequency scale in Hz. The vertical ruler represents the relative frequency and the horizontal axis represents the duration of the sound example. Due to software limitations and overlapping elements, the amplitude level of the waveform (between −1 and +1) and the time ruler are not displayed.Spectrogram: The second layer shows the frequency distribution and intensity over time, highlighting harmonic structures and temporal changes. The spectrogram was calculated using a 1024-point FFT analysis window (in greenish background) on a Mel scale. A larger 4096-point FFT window (in sunset colors) is superimposed, highlighting the highest intensity partials of the sound spectrum.Phase Visualization: The third layer shows the phase of the sound wave partials, giving an insight into the evolution of the sound wave components. The phase variation has been analyzed using a short 512-point FFT window on a linear phase analysis scale. This layer visualizes the interaction between the phases of the partials and shows how phase alignment or misalignment affects the overall sound wave. Constructive interference results in the amplification of certain frequencies, while destructive interference can cause phase cancellation, altering the perceived tonal quality of the sound. Regular patterns appear when there is phase alignment between partials.Instantaneous Spectral Content: The fourth layer displays a point on the spectral content, providing a detailed snapshot of the frequency peak of the sound at a given point in time. The instantaneous spectral analysis was performed on the 250 Hz point of the spectrum, indicated by a blue marker on the frequency ruler. The horizontal axis represents the spectral frequency at a given time, while the vertical axis shows the relative intensity of spectral regions, with their values displayed on the vertical axis.

### 5.1. Sound Colour Attributes

Table 2 shows the percentage of participants who associated specific color-related descriptors—Bright, Dull, Dark, Clarity, and Other—with each of the 18 abstract sounds analyzed in the SEQR survey. The percentages for each descriptor in a row sum to 100% and reflect the relative frequency with which each descriptor was selected by participants to characterize each sound. This allows us to examine the dominant perceptual associations for each sound, providing insights into how listeners interpret and categorize sound color based on subjective qualities. The distribution of these descriptors provides a quantitative view of auditory perception and serves as a basis for understanding the relationships between sound attributes and listeners’ subjective interpretations.

**Table 2 behavsci-15-00396-t002:** Participant selections of color-related descriptors for each abstract sound. Percentages for each descriptor in each row sum to 100%.

Colour					
Sound	Bright	Dull	Dark	Clarity	Other
**#1**	24.1	27.8	3.7	40.7	3.7
**#2**	10.0	45.0	35.0	10.0	—
**#3**	8.1	45.9	29.7	10.8	5.4
**#4**	3.3	36.7	50.0	10.0	—
**#5**	8.3	33.3	8.3	50.0	—
**#6**	52.2	21.7	4.3	21.7	—
**#7**	4.2	54.2	25.0	16.7	—
**#8**	9.5	42.9	28.6	19.0	—
**#9**	38.1	23.8	9.5	28.6	—
**#10**	55.0	—	10.0	35.0	—
**#11**	11.1	16.7	50.0	16.7	5.6
**#12**	57.9	10.5	5.3	26.3	—
**#13**	11.1	50.0	16.7	22.2	—
**#14**	47.1	11.8	—	41.2	—
**#15**	17.6	29.4	17.6	35.3	—
**#16**	18.8	18.8	31.3	31.3	—
**#17**	18.8	31.3	25.0	25.0	—
**#18**	29.4	47.1	5.9	17.6	—

A thorough examination of the data reveals discernible perceptual tendencies across the different sound categories. In many cases, the attribute Dull emerges as the most frequently selected descriptor, particularly for Sounds #2, #3, #4, #7, #8, #13, and #18. This prevalence suggests a dominant perception of low brightness or dull characteristics in these sound profiles. In contrast, the attribute Bright was perceived as more intense in Sounds #12 (57.9%), #10 (55%), #6 (52.2%), and #14 (47.1%), highlighting a strong perceptual emphasis on higher frequencies or more resonant qualities in these cases.

The attribute Clarity also stands out as a notable feature in some of the sound profiles. Sound #5 achieves the highest percentage of clarity at 50%, closely followed by Sound #14 (41.2%) and Sound #1 (40.7%). These results suggest that these sounds are perceived as having distinct or well-defined characteristics, probably influenced by their spectral composition. Furthermore, a positive correlation is observed between high levels of brightness and clarity, as exemplified by sounds #12 and #10. However, Sound #5, with a clarity of 50.0% and a brightness of only 8.3%, shows that this is not an absolute rule. This highlights the complexity of auditory perception, where factors beyond brightness—such as texture-specific details—can influence the interpretation of these attributes.

#### 5.1.1. Sound #16 and Sound #17

The color perception scores for Sounds #16 and #17 show that both are characterized by relatively balanced descriptors, with a slight emphasis on Dark and Dull qualities and moderate Clarity. The lack of a significant association with Bright attributes suggests a preponderance of low- to mid-frequency components or less defined high-frequency energy. The results of the color perception analysis for Sounds #16 and #17, as shown in Figure 4, are remarkably similar, with the distributions across the Bright, Dull, Dark, and Clarity categories all ranging from 18.8% to 31.3%.

The similarity in the synthesis techniques used to create the two sounds may explain the consistency in the descriptors observed. However, a closer examination of their distinctive sound wave characteristics reveals notable discrepancies. These include variations in spectral peaks, linear centroids, and the evolution of the sound wave (see Figure 4).

Sound #16 displays irregular fluctuations in amplitude over its duration, suggesting a more complex or dynamic sound structure. While the frequency centroid remains relatively stable with small deviations, indicating moderate fluctuations in the frequency content of the sound over time, the spectral content appears to be dispersed. The spectrogram shows a wide frequency range with some high intensity partials in the mid to high frequencies. The instantaneous spectrum analysis shows more energy peaks, particularly in the higher frequencies, above 2000 Hz. This presence of higher frequency spectral content could be associated with perceptual descriptors such as Bright and Clear.Sound #17, however, shows a more stable and repetitive pattern compared to #16, thus suggesting a less complex or more regular internal structure. The frequency centroid shows more consistent behavior, indicating that the spectral content of the sound is relatively stable over time. The spectrogram displays a broad frequency spectrum, with an obvious dominance in the middle region of the spectrum, as also confirmed by the instantaneous spectrum analysis. The strongest energy distribution is also observed in the mid frequencies, particularly between 1200 Hz and 1800 Hz, suggesting a reduced brightness compared to #16. This feature contributes to the Dull descriptor.

In conclusion, Sound #16 is characterized by brightness and clarity, while #17 tends towards dullness and darkness due to its lower frequency dominance and stable, less dynamic structure.

Given these results, which perceptual factors might explain the similarity of the participants’ descriptions of these sounds? It is possible that the common synthesis technique used to create the sounds induced common features, particularly in the more energetic regions of the spectrum. These common features could explain why participants gave similar descriptions. However, the lack of a dominant perceptual tendency in either sound suggests that listeners encountered ambiguous auditory qualities that were open to multiple interpretations.

#### 5.1.2. Sound #2 and Sound #3

The results for Sounds #2 and #3 also show comparable color perceptions, characterized by increased values for Dull and Dark and decreased values for Bright and Clarity. However, in contrast to Sounds #16 and #17, there is a more evident correlation between the intrinsic characteristics of the sound waves and the participants’ descriptions. The Dark and Dull descriptors together account for between 75.5% and 80% of the responses, indicating that these sounds were perceived as less ambiguous than the previous pair.

Despite their different production techniques, both sounds exhibit a predominance of spectral energy below 2500 Hz, as shown in Figure 5, with more pronounced peaks observed between 250 Hz and 750 Hz. In addition, both sounds show a rich concentration of partials above 1300 Hz, although with slightly less energy. This similarity in spectral organization may provide a rationale for the observation that participants assigned high values to the descriptors Dull and Dark, suggesting that the concentration of lower-frequency components contributes significantly to these descriptors. Examples such as these support the idea that color attributes such as bright, dull, dark, or clarity are independent of timbre, at least in the traditional instrumental sense.

Sound #2 was created using additive synthesis, resulting in a near-harmonic spectrum with distinct spectral peaks regularly distributed across the spectrum. The frequency centroid shows stability with minor fluctuations, indicating that the sound maintains a consistent spectral content over time. The spectrogram shows moderate energy distributed across the low-mid region of the spectrum, and the instantaneous spectrum confirms a steady intensity in the low-mid frequencies. The phase plot shows the expected regularity of an additive synthesized sound wave.Sound #3, produced by a noise generator with light filtering, has a more diffuse spectral profile, as expected from a noisy sound wave. Despite this, the auditory descriptors are similar to those of Sound #2. The waveform shows more fluctuation, indicating a more dynamic or complex sound, and the frequency centroid shows a more irregular frequency range. The spectrogram shows a concentration of partials in the lower frequency range, and the instantaneous spectrum analysis supports this by showing intensity primarily in the lower frequencies, reinforcing the dullness and darkness of the sound.

These observations seem to indicate that sounds characterized by a spectral concentration of partials in the lower region of the spectrum—whether produced with more harmonic structures or more chaotic, noisy ones—tend to be perceived as darker and duller. Although spectral analysis suggests that these sounds would be audibly different, participants’ responses suggest that their perception of color attributes may transcend specific sound wave properties.

#### 5.1.3. Sound #10 and Sound #4

As shown in Figure 6, Sounds #10 and #4 exhibit contrasting color perceptions, despite sharing some similarities in their sound structures.

Sound #10 is characterized by elevated Bright (55%) and Clarity (35%) values, combined with minimal Dark (10%) and an absence of Dull descriptors. In contrast, Sound #4 has a distinctly different profile, with only 3.3% Bright and 10% Clarity, but is perceived predominantly as Dark (50%) and Dull (36.7%).

Sound #10 shows irregular fluctuations in amplitude, and while the frequency centroid shows some variation, it remains generally stable. This indicates a relatively consistent distribution of spectral content over the duration of the sound. The spectrogram shows a significant number of partials in the mid frequencies, which is confirmed by the instantaneous spectrum analysis, which shows a relatively even distribution of energy in the lower frequencies, with slightly more energy in the higher frequencies. These characteristics may explain why Sound #10 is perceived as bright and clear.Sound #4, however, has a more variable waveform compared to #10, suggesting a more complex sound. The frequency centroid in Sound #4 is more consistent over time, with fewer fluctuations which could indicate a sound with a more defined pitch. The spectral content distribution is centered in the lower frequencies, with a clear peak around 250 Hz, as shown in the instantaneous spectrum analysis.

An examination of the internal structure of these sounds reveals that Sound #10 concentrates its spectral energy in the higher frequency range, while Sound #4 has a predominant low-frequency spectral energy distribution (Figure 6). This distribution supports the conclusion that, in a manner analogous to instrumental sounds, abstract and complex sounds with strong energy in the higher regions of the spectrum tend to evoke perceptions of brightness, while strong energy in the lower regions of the spectrum contributes to perceptions of dullness or darkness.

#### 5.1.4. Sound #12 vs. Sound #7

As shown in Figure 7, while both Sound #12 and Sound #7 exhibit regular and stable structures, they differ significantly in their spectral composition, leading to distinct perceptual experiences.

A comparison of Sounds #12 and #7 reveals a stark contrast in their color perceptions. Sound #12, with 57.9% Bright and 26.3% Clarity, low Dark (5.3%) and Dull (10.5%) values, contrasts sharply with Sound #7 (4.2% Bright and 16.7% Clarity; 54.2% Dull and 25.0% Dark), which has almost opposite distribution patterns for the color descriptors.

Sound #12, produced by granular synthesis, has spectral components concentrated in a relatively narrow and stable frequency range, as shown in the spectrogram. The waveform appears relatively uniform, with a frequency centroid that remains stable, indicating that the sound’s spectral content of the sound is consistent throughout its duration. The instantaneous spectrum confirms the concentration of energy in a narrow frequency band between 700 Hz and 1300 Hz.Sound #7, created by additive synthesis, has spectral components that are more evenly distributed across the spectrum, with partials of similar intensity. The waveform shows a repetitive pattern with a stable frequency center. The spectrogram shows a broad frequency spectrum, with energy present at similar intensities throughout the spectrum up to about 3200 Hz, where it drops off abruptly.

The perceived color difference appears to be related to the position and range of the spectral components rather than the overall spectral distribution (Figure 7). The concentration of mid-range components in Sound #12 contributes to its perception as brighter, whereas the more even distribution of partials in Sound #7, which extends beyond Sound #12 into higher frequencies, is perceived as darker and duller.

Although it may seem counterintuitive that a sound with a more regular spectral distribution can be perceived as darker, this suggests that the concentration of energy in specific frequency regions plays a more significant role in the perception of brightness or dullness than the overall regularity of the spectral components. It also highlights the sensitivity of our perception to mid-to-low frequencies, which may influence these results.

### 5.2. Sound Texture Attributes

Sound texture refers to the qualitative characteristics of a sound that result from its spectral structure, shaped by the distribution and evolution of the sound wave’s partials, which together influence how a sound is perceived. These characteristics are grouped into five different categories: Smooth, Rough, Granular, Striated, and Iterative. As shown in Table 3, Sound #1 is the only sound with a strong association (88.5%) with the Smooth attribute, a result attributed to its sinusoidal profile. It is noteworthy that this sound is the only one with such a strong association with smoothness, with a minimal presence in the other categories.

**Table 3 behavsci-15-00396-t003:** Participant selections of texture-related descriptors for each abstract sound. Percentages for each descriptor in each row sum to 100%.

Texture						
Sound	Smooth	Rough	Granular	Striated	Iterative	Other
#1	88.5	3.8	—	5.8	1.9	—
#2	21.1	23.7	7.9	44.7	—	2.6
#3	2.6	34.2	55.3	2.6	5.3	—
#4	3.3	50.0	16.7	16.7	6.7	6.7
#5	4.2	20.8	37.5	29.2	8.3	—
#6	17.4	34.8	8.7	34.8	4.3	—
#7	16.7	50.0	25.0	8.3	—	—
#8	22.7	22.7	22.7	22.7	9.1	—
#9	19.0	9.5	23.8	38.1	9.5	—
#10	10.0	5.0	50.0	30.0	5.0	—
#11	27.8	16.7	11.1	16.7	22.2	5.6
#12	31.6	10.5	15.8	26.3	15.8	—
#13	27.8	16.7	—	33.3	22.2	—
#14	11.8	35.3	11.8	23.5	17.6	—
#15	6.3	25.0	31.3	18.8	18.8	—
#16	12.5	37.5	25.0	12.5	6.3	6.3
#17	12.5	6.3	56.3	18.8	6.3	—
#18	—	47.1	17.6	29.4	5.9	—

The Smooth texture is also dominant in Sounds #11 and #12, where higher percentages indicate a perception of smoothness. However, these sounds also achieve significant percentages in other texture attributes, suggesting a more ambiguous or layered texture. This suggests that specific spectral evolutions within these sounds may influence listeners’ perceptions of smoothness, even when synthetic textures exhibit greater complexity.

Sound #8 has a balanced distribution across the Smooth, Rough, Granular, and Striated categories (each at around 22.7%), indicating a perception of layered or nuanced texture without a dominant quality. Similarly, Sounds #14, #15, and #16 showed strong tendencies to be perceived as Rough or Granular, with Striated qualities appearing as secondary characteristics.

The largest overall percentages were assigned to the Rough, Granular, and Striated categories. It is noteworthy, however, that certain sounds exhibit a distinct profile with respect to these attributes. Specifically, Sounds #4, #7, and #18 were predominantly classified as Rough; Sounds #3, #10, and #17 were predominantly classified as Granular; and Sound #2 was predominantly classified as Striated.

The Iterative texture descriptor is less prominent across the dataset, suggesting that it functions more as a secondary quality. This is likely due to the tendency for sounds with rhythmic or repetitive elements to be classified as Granular or Striated due to their inherent complexity, making the Iterative quality less prominent as a primary descriptor.

In terms of texture perception, Sounds #3, #10, and #17 show a high degree of similarity, particularly in the Granular texture category, with percentages of 55.3%, 50.0%, and 56.3%, respectively. However, there is a divergence in the rating of the Rough and Striated attributes:Sound #3 was classified with a significantly higher rating in the Rough attribute (34.2%) compared to Sounds #10 and #17, which show significantly lower ratings of 5.0% and 6.3%, respectively.Sounds #10 and #17 have a significant amount of Striated texture (30.0% and 18.8%), while Sound #3 has only 2.6% in this category.

This variation suggests that while granularity is a common attribute, the perception of roughness and striation differs between the sounds. The perceived roughness of Sound #3 (34.2%) can be attributed to the greater irregularity within the sound wave, which has no discernible regular pattern (Figure 5). In addition, the spectrum of Sound #3 contains a greater proportion of low frequency content, despite the presence of spectral peaks distributed across the frequency range. In contrast, Sounds #10 and #17 exhibit spectral weight in the mid-to-high frequencies, and regular spectral patterns are visible in the spectrogram that are likely responsible for the striated quality observed (see Figure 6 for #10 and Figure 4 for #17). Sound #17 has a distinct repetitive pattern in its sound wave, which may account for the increased level of striated quality.

#### 5.2.1. Sound #11 and Sound #13

Sounds #11 and #13 have identical values for Smooth, Rough, and Iterative, at 27.8%, 16.7%, and 22.2%, respectively. The main differences are in granularity and striation. Although both sounds were produced using additive-based sound synthesis, their similarity in participants’ responses contrasts with significant differences in the physical properties of their sound waves, as can be seen in Figure 8.

Sound #13, despite its near-harmonic spectrum, exhibits a deep tremolo-like vibration with prominent spectral peaks at 350 Hz, 650 Hz, and 950 Hz, giving it a mid-range perception. The tremolo introduces same variations in the frequency centroid, contributing to the irregularity of the perceived texture. The spectrogram and phase analysis show a relatively stable pattern, with clear synchrony between the partial phases below 1200 Hz, indicating a striated texture. While the spectral peaks are close to harmonic, the regular fluctuations in waveform and phase further support the perception of striation.In contrast, Sound #11 has a lower register profile with spectral partials concentrated below 400 Hz and a more stable frequency centroid. The spectrogram for Sound #11 shows a more even distribution of energy across the lower frequencies, which is further supported by the less differentiated peaks observed in the instantaneous spectrum analysis. In addition, the stable spectral structure seen in the phase analysis layer confirms a consistent internal structure.

Thus, the differences in internal structure patterns between Sounds #11 and #13 are influenced by their frequency register and spectral stability, which in turn influence the perception of texture. Despite their different spectral patterns, both sounds produce similar perceptual sensations. Despite the apparent differences in their physical properties, the auditory perception of these sounds is more closely aligned in terms of granularity and striation. This highlights the complexity of auditory perception and shows that perceived texture can be influenced by factors other than spectral properties alone.

#### 5.2.2. Sound #14 and Sound #15

As can be seen in Figure 9, the distribution of the texture descriptors over Sounds #14 and #15 is relatively even for both, with no category exceeding 31.3%. It is worth noting that both sounds are derived from natural recordings, but have undergone different processing, resulting in distinctly different sound waves.

The natural sound that gave rise to Sound #14 has undergone extensive processing to produce a medium-high buzzing sound with a more structured and repetitive profile. The waveform has noticeably sharp peaks, while the frequency centroid shows some stability. The spectrogram shows a clear concentration of energy in the mid-to-high frequency range with repetitive spectral patterns, reinforcing the perception of a striated and structured texture that contributes to the buzzing auditory sensation.Sound #15, also generated from a natural source that has been lightly processed, retains an irregular, liquid-like quality with random, grainy textures resembling large falling water droplets. The waveform shows irregularity with sharp fluctuations, indicating an unpredictable or random structure. The frequency centroid shows greater variation with more pronounced shifts, suggesting a more complex or irregular spectral content. The spectrogram shows clear energy peaks at both low and high frequencies. However, the distribution is more spread out compared to Sound #14. These fluctuations, or seemingly erratic behavior over time, contribute to the perceived granularity or roughness. In addition, the instantaneous spectrum shows a more irregular distribution of energy peaks.

Despite the differences in spectral content and sound wave characteristics that are clearly visible in Figure 9, the balanced distribution of responses suggests that listeners perceive these sounds as texturally complex, with no single dominant attribute. The lack of a discernible texture profile suggests that Sounds #14 and #15 are perceived as indistinct, with overlapping or intertwined textures rather than a dominant quality. This complexity is probably due to the inherent specificity of the natural sounds and the processing techniques used, which increase the perceived depth and variety of the textures.

### 5.3. Integrating Colour and Texture Attributes

In this section we integrate the perceptual results of the color and texture attributes from the survey with the physical analysis of the sound waves. This combined approach allows for a more comprehensive understanding of sound similarities by considering both texture and color simultaneously.

Differences between sounds were quantified using the Euclidean distance, which measures the similarity between sounds based on their combined attributes. The resulting values reflect relative perceptual distances, with smaller values indicating greater similarity and larger values indicating greater difference. These distances are calculated by taking the square root of the sum of the squared differences between the attributes of each sound pair. The integrated Euclidean distances shown in Table 4 were derived from the percentage values assigned to texture and color. Each individual Euclidean distance (for texture and color attributes) was calculated as the absolute difference between the percentage values assigned to that attribute for each sound. For example, if Sound #5 has a texture value of “Smooth” of 4.2% and Sound #15 has a value of “Smooth” of 6.3%, the resulting Euclidean distance for this attribute is:∣4.2 − 6.3∣ = 2.1

This method is applied to all texture and color attributes, providing a simple measure of similarity for each perceptual dimension. Once the individual attribute Euclidean distances have been calculated, the total Euclidean distance between two sounds is determined by summing the squares of these values and taking the square root of the total.

For example, in Sound #5 and Sound #15, the individual attribute distances shown in Table 5 below were calculated using the absolute differences between the percentage values assigned to each texture and color attribute. The total Euclidean distance between Sound #5 and Sound #15 is then determined by summing the squares of these values and taking the square root:$$\sqrt{{\left(2.1\right)}^{2}+{\left(4.2\right)}^{2}+{\left(6.2\right)}^{2}+{\left(10.4\right)}^{2}+{\left(10.5\right)}^{2}+{\left(9.3\right)}^{2}+{\left(9.3\right)}^{2}+{\left(14.7\right)}^{2}}=36.8$$

This procedure is applied to all sound pairs and allows a quantitative assessment of perceptual similarity when both color and texture are considered together.

The relationships between the sounds, as quantified by the combined Euclidean distances, are presented in Table 4. This methodological framework allows us to identify sound pairs that are perceived as highly similar, as well as those that show subtle differences in terms of both color and texture attributes.

As illustrated in the above table, the grey cell highlights the largest and smallest Euclidean distances. The four additional highlighted values correspond to the sound pairs to be discussed in the following section.

It might be assumed that two sounds that are objectively similar, based on their production method or source, would also be perceived as similar in color and texture. However, the perceptual data do not always support with this assumption. Conversely, sounds that differ significantly in their origin, synthesis method, or spectral properties can sometimes be perceived as more similar, contrary to the expected patterns.

**Table 4 behavsci-15-00396-t004:** Combined Euclidean distances of color and texture attributes between 18 sounds, highlighting the largest, smallest, and four selected sound pairs. Cells highlighted in grey indicate the minimum and maximum distances; cells highlighted in blue indicate the four selected sound pairs discussed below.

Sound	#1	#2	#3	#4	#5	#6	#7	#8	#9	#10	#11	#12	#13	#14	#15	#16	#17	#18
**#1**	—																	
**#2**	129.5	—																
**#3**	137.7	72.6	—															
**#4**	**157.6**	61.7	66.9	—														
**#5**	109.0	87.1	81.3	96.1	—													
**#6**	117.3	73.7	116.0	98.3	89.0	—												
**#7**	131.5	64.5	49.3	49.1	82.9	96.4	—											
**#8**	110.0	37.8	53.8	59.5	63.0	76.8	**44.7**	—										
**#9**	97.0	69.7	102.1	107.0	62.7	46.8	99.9	61.1	—									
**#10**	114.3	106.8	97.4	127.9	70.3	66.4	112.3	87.6	47.6	—								
**#11**	121.1	62.5	94.3	67.0	96.2	94.9	84.3	54.1	80.8	111.5	—							
**#12**	103.1	94.2	127.9	127.4	96.9	45.8	119.4	81.4	44.4	52.7	80.1	—						
**#13**	105.1	37.5	64.6	85.8	61.6	79.0	55.9	**32.9**	56.6	73.8	**64.0**	74.4	—					
**#14**	114.3	84.9	110.7	81.1	77.6	**41.1**	90.5	75.3	53.4	60.9	69.7	50.6	82.6	—				
**#15**	103.9	73.2	66.8	75.7	**36.8**	74.8	64.3	46.6	54.0	71.0	70.9	81.1	52.7	58.1	—			
**#16**	114.5	75.0	69.2	55.0	62.7	72.4	55.3	49.7	68.3	86.8	58.7	86.1	73.2	51.5	38.6	—		
**#17**	104.5	82.5	57.3	93.2	59.2	98.6	83.3	55.5	64.7	53.7	82.5	95.1	50.4	92.5	47.1	60.0	—	
**#18**	80.1	65.4	80.7	65.9	74.2	50.6	55.0	56.5	66.7	84.4	93.9	85.3	56.7	64.6	61.0	62.6	85.2	—
**MIN**	**80.1**	**37.5**	**49.3**	**49.1**	**36.8**	**41.1**	**44.7**	**32.9**	**44.4**	**52.7**	**58.7**	**50.6**	**50.4**	**51.5**	**38.6**	**60.0**	**85.2**	**—**
**MAX**	**157.6**	**106.8**	**127.9**	**127.9**	**96.9**	**98.6**	**119.4**	**87.6**	**80.8**	**111.5**	**93.9**	**95.1**	**82.6**	**92.5**	**61.0**	**62.6**	**85.2**	**—**

As shown in the table, Sounds #8 and #13 are the most similar perceptually, although they are objectively different. Sound #8 has a buzzing quality, is perceived mainly in the middle register, and contains a slight amplitude modulation. In contrast, Sound #13 has a slow, deep tremolo-like oscillation in a near-harmonic spectrum, also in the middle register, but with a slightly higher pitch. Both sounds were created using additive-based hybrid synthesis, a method which may offer a rational explanation for their perceptual similarity.

For example, Sounds #7 and #8 share identical synthesis parameters, both generated by additive-based hybrid synthesis in the mid-to-low frequency range, with only a slight amplitude modulation applied to Sound #8. Despite these similarities, they are perceived as being similar in color but very different in texture. Similarly, Sounds #5 and #15 were both produced using granular synthesis applied to natural sound sources, with the spectral content centered in the mid-to-low register. Although these sounds have similar textures, they differ in perceived color. These examples suggest that perceptual similarity is influenced not only by the production method and spectral properties, but also by the interaction between internal spectral content (influencing color perception) and temporal evolution (influencing texture perception). The evolution of these characteristics over time can have a significant impact on how sounds are perceived, sometimes leading to perceptual results that contradict objective acoustic similarities.

To better understand these discrepancies, perceptual similarities are examined by combining the texture and color descriptors for selected sound pairs using comparative tables of Euclidean distances. Each row in the table represents one of the nine attributes for each sound pair, and the individual Euclidean distance indicates where the sounds are aligned (low distance) or diverge (high distance). As previously described, the Euclidean distance for each individual attribute is calculated by taking the absolute difference between the percentage values assigned to that attribute for each sound.

Graphical representations of the survey results are used to illustrate how participants responded to each sound. Each figure contains two graphs: one showing the perceived color attributes (on the left) and the other showing the texture attributes (on the right). The lower part of the figure shows the physical properties of each sound across the frequency spectrum, which helps to link the perceived qualities to their corresponding spectral characteristics. By combining the sounds in different pairs, the visualization reveals relationships that may not be immediately obvious when analyzing each attribute separately.

#### 5.3.1. Integrated Attributes of Sound #5 and Sound #15

As shown in Table 5, a comparison of the combined texture and color attributes for Sounds #5 and #15 reveals remarkable perceptual similarities.

**Table 5 behavsci-15-00396-t005:** Euclidean distance between Sound #5 and Sound #15 attributes.

Attribute	#5 Value	#15 Value	Euclidean Distance
**Texture**			
**Smooth**	4.2	6.3	2.1
**Rough**	20.8	25.0	4.2
**Granular**	37.5	31.3	6.2
**Striated**	29.2	18.8	10.4
**Iterative**	8.3	18.8	10.5
**Color**			
**Bright**	8.3	17.6	9.3
**Dull**	33.3	29.4	3.9
**Dark**	8.3	17.6	9.3
**Clarity**	50.0	35.3	14.7

In terms of texture, the Euclidean distances are minimal for the Smooth, Rough, and Granular attributes. Of these, the Smooth attribute has the smallest difference, with a value of only 2.1%. This suggests that both sounds have a similar textural quality.

In terms of color, Sounds #5 and #15 show a high degree of similarity, with a minimal Euclidean difference of only 3.9% in Dullness. Both sounds are primarily perceived as dull and clear, although slight discrepancies in their Brightness and Darkness values account for a difference of 9.3%. Specifically, Sound #5 displays Brightness and Darkness values of 8.3%, while Sound #15 displays slightly higher values of 17.6%. These moderate differences reflect variations in their respective frequency distributions.

The most noticeable differences are observed in the Clarity and Dullness attributes. Specifically, the Clarity attribute shows a moderate discrepancy, with Sound #5 reaching 50.0% clarity and Sound #15 reaching 35.3%. This indicates that Sound #5 is perceived as slightly clearer, a result that can be attributed to the stability and more energetic high frequency partials in Sound #5, as evidenced by its frequency centroid (see Figure 10).

While Sound #5 is perceived as clearer and slightly less bright and dark than Sound #15, there are more pronounced differences in their textural qualities. Notably significant are the discrepancies observed in the Striated and Iterative attributes. Specifically, Sound #5 has 29.2% Striated and 8.3% Iterative, while Sound #15 has 18.8% for both attributes. These differences contribute to subtle textural contrasts between the sounds. Spectral analysis shows that the spectral content of Sound #5 is more regular, while Sound #15 has a more fragmented and dynamic texture, with greater irregularity in its spectral structure over time.

Spectral analysis also reveals a slightly wider spectral peak band for Sound #5, ranging from 700 Hz to 1100 Hz, compared to Sound #15’s narrower peak range of 600 Hz to 850 Hz. Despite observable differences in their waveforms and dynamic profiles (see Figure 10), both sounds share similarities in their spectral peaks and exhibit irregular shifts in the frequency centroid. These common characteristics, together with the periodicity observed in their dynamic profiles, contribute to the overall similarity in both texture and color perception.

#### 5.3.2. Integrated Attributes of Sound #6 and Sound #14

As shown in Table 6, Sounds #6 and #14 were perceived as similar in both color and texture attributes, with minimal differences in Roughness (0.5%) and Granularity (3.1%). These small differences suggest that both sounds have a similar rough, grainy texture. This is further supported by the comparable Brightness attribute (Euclidean difference of 5.1%), indicating that both sounds are perceived as relatively bright.

**Table 6 behavsci-15-00396-t006:** Euclidean distance between Sound #6 and Sound #14 attributes.

Attribute	#6 Value	#14 Value	Euclidean Distance
**Texture**			
**Smooth**	17.4	11.8	5.6
**Rough**	34.8	35.3	0.5
**Granular**	8.7	11.8	3.1
**Striated**	34.8	23.5	11.3
**Iterative**	4.3	17.6	13.3
**Color**			
**Bright**	52.2	47.1	5.1
**Dull**	21.7	11.8	9.9
**Dark**	4.3	—	4.3
**Clarity**	21.7	41.2	19.5

The similarities between the two sounds are further emphasized by small differences in smoothness and darkness. Sound #6 is perceived as slightly smoother (with a Euclidean difference of 5.6%) and darker (with a darkness score of 4.3%) than Sound #14, which has no perceived darkness. These differences can be attributed to the more prominent low frequency partials in Sound #6, as shown in Figure 11.

Despite these differences, the two sounds have a similar texture in terms of roughness and granularity. However, a more noticeable difference is observed in the Clarity attribute, where Sound #14 is perceived as significantly clearer (41.2%) compared to Sound #6 (21.7%). This difference is probably due to the rougher, more granular nature of Sound #14, which results in less tonal clarity than Sound #6.

In terms of texture, there is a significant difference between the two sounds. Sound #6 has lower ratings in the Striated (11.3%) and Iterative (13.3%) qualities compared to Sound #14. Sound #6 is perceived as more stable and less structured, while Sound #14 has more irregularities in its waveform and phase patterns, contributing to its more complex texture.

In terms of spectral characteristics, Sound #6 has more pronounced low-to-mid frequency peaks, which probably contributes to its perception as a duller and slightly darker sound. In contrast, although Sound #14 has low-frequency components, it has less intensity in these areas, resulting in a perception of greater clarity and brightness. However, both sounds have a comparable spectral density above 1500 Hz, as observed in the instantaneous spectrum analysis, and show similar behavior in their linear frequency centroids. These spectral similarities may explain the overall perceptual match between the two sounds, even though Sound #6 is synthetic and Sound #14 is based on a natural recording (Figure 11).

#### 5.3.3. Integrated Attributes of Sound #7 and Sound #8

The production of Sounds #7 and #8 is quite similar, both featuring a medium-low-pitch buzzing sound and having the same spectral centroid around 1800 Hz. Sound #8 also contains a slight amplitude modulation. This similarity is reflected in their perceptual classifications (see Table 7), with minor Euclidean differences in some attributes, suggesting that both sounds have almost identical perceived textural structures.

**Table 7 behavsci-15-00396-t007:** Euclidean distance between Sound #7 and Sound #8 attributes.

Attribute	#7 Value	#8 Value	Euclidean Distance
**Texture**			
**Smooth**	16.7	22.7	6.0
**Rough**	50.0	22.7	27.3
**Granular**	25.0	22.7	2.3
**Striated**	8.3	22.7	14.4
**Iterative**	—	9.1	9.1
**Color**			
**Bright**	4.2	9.5	5.3
**Dull**	54.2	42.9	11.3
**Dark**	25.0	28.6	3.6
**Clarity**	16.7	19.0	2.3

For example, the Euclidean distance for granularity is only 2.3% and for smoothness it is 6%. Both sounds also have similar levels of darkness, with only a 3.6% difference, and a modest 2.3% difference in Clarity, reinforcing their perceptual similarity. This is further supported by the spectrograms: Sound #7, perceived as Dull (54.2%) and Dark (25.0%), has a broader energy distribution below 3200 Hz compared to Sound #8.

On the other hand, Sound #8, with amplitude modulation applied, draws the listener’s attention to the middle register, resulting in a slightly higher perceived Brightness (9.5%) and Clarity (19.0%) compared to Sound #7, with values of 4.2% and 16.7%, respectively. Sound #8 also has a slightly lower percentage of Darkness (43% for Sound #8 vs. 54% for Sound #7). This suggests that the modulation process affects color perception by emphasizing the mid-range frequencies.

Minor discrepancies are observed in the Smoothness (22.7% for Sound #8 vs. 16.7% for Sound #7) and Brightness (9.5% for Sound #8 vs. 4.2% for Sound #7) categories. However, the most significant difference is in Roughness, where Sound #7 is rated at 50.0% in comparison to 22.7% for Sound #8. Visual analysis of the spectrograms shows that both sounds have broad frequency distributions with energy concentrated in the mid- and low-frequency ranges. While both sounds have spectral peaks, the distribution of these peaks differs slightly in terms of intensity and concentration.

Sound #8 shows a more balanced perception across the texture categories, with ratings of 22.7% for Smooth, Rough, Granular, and Striated textures. This suggests that its texture is perceived as more layered or ambiguous. In contrast, Sound #7 is primarily characterized by roughness and granularity. Despite their similar production methods, sound wave structure (Figure 12), and comparable granularity and clarity, Sound #8 is perceived as smoother and more layered, while Sound #7 is perceived as rougher and more straightforward in texture.

#### 5.3.4. Integrated Attributes of Sound #11 and Sound #13

As shown in Table 8, Sounds #11 and #13 show identical values in the Smooth, Rough, and Iterative categories, suggesting a similar textural quality. They also share equivalent levels of Brightness and demonstrate comparable Clarity, with a minimum Euclidean distance of 5.5%. In addition, both sounds exhibit similar granularity, with only an 11.1% difference between them.

These common attributes suggest a similar overall texture and perceptual clarity, despite differences in specific textural qualities. However, closer analysis of their sound waves reveals distinct spectral differences.

**Table 8 behavsci-15-00396-t008:** Euclidean distance Between Sound #11 and Sound #13 attributes.

Attribute	#11 Value	#13 Value	Euclidean Distance
**Texture**			
**Smooth**	27.8	27.8	—
**Rough**	16.7	16.7	—
**Granular**	11.1	—	11.1
**Striated**	16.7	33.3	16.6
**Iterative**	22.2	22.2	—
**Color**			
**Bright**	11.1	11.1	—
**Dull**	16.7	50.0	33.4
**Dark**	50.0	16.7	33.4
**Clarity**	16.7	22.2	5.5

These common attributes suggest a similar overall texture and perceptual clarity, despite differences in specific textural qualities. However, closer analysis of their sound waves reveals distinct spectral differences. Spectral analysis, instantaneous spectral analysis peaks, and phase analysis show significant differences between the two sounds (see Figure 13). These differences highlight the fact that while the sounds may be similar in some respects, their perceptual qualities are shaped by different spectral content.

In terms of texture, the most striking difference is in the Striated attribute, where Sound #13 has a significantly higher value (33.3% vs. 16.7%), indicating a more clearly striated and structured texture compared to Sound #11. This difference is further emphasized by phase and spectrogram analyses, which reveal more defined spectral patterns and greater energy concentration at regular intervals for Sound #13 (see Figure 13). The increased striations and dynamic fluctuations in Sound #13 result in a more complex, layered texture, while Sound #11 maintains an overall smoother and more stable texture.

The most pronounced differences between the two sounds are in their Dull and Dark attributes. Sound #13 is significantly duller (50.0% vs. 16.7%), giving it a softer, more subdued quality. Sound #11, on the other hand, is characterized by a deeper, more pronounced low-register quality, as indicated by its higher Darkness rating (50.0% vs. 16.7%). This difference can be attributed to the concentration of low frequency energy in Sound #11, as shown by its spectrogram and instantaneous spectral analysis peaks. The lower register content of Sound #11 contributes to its darker, grounded texture, while the higher register focus of Sound #13 provides a more dynamic, varied tonal quality.

As Figure 13 shows, Sound #11 has a lower register profile with spectral partials concentrated below 400 Hz, while Sound #13 has a nearly harmonic structure with prominent spectral peaks at 350 Hz, 650 Hz, and 950 Hz. These spectral differences give rise to perceptual contrasts: Sound #11 feels more stable and grounded in the lower register, while Sound #13 has a tremolo-like dynamic quality perceived in the mid-range.

This analysis highlights that comparable distributions across texture categories may indicate perceptual ambiguity rather than simple similarity. It underlines the role of register and spectral regularity in shaping texture perception. The striation and roughness of Sound #13 reflect a more dynamic and irregular internal structure, whereas the clarity and darkness of Sound #11 are associated with a more stable, low-frequency profile. Ultimately, these results demonstrate that subtle changes in spectral content and frequency distribution can significantly influence how texture is perceived.

## 6. Discussion of Results

The analysis of the SEQR survey results, with a particular focus on color and texture qualities, reveals the complex relationships between intrinsic sound characteristics. The data substantiate the hypothesis that participants employ multisensory descriptors to refer to abstract sounds. The examination of the correlations between subjective descriptors and objective sound properties has given rise to the emergence of patterns, which have suggested the interplay between spectral properties and perceptual outcomes.

### 6.1. Correlations Between Descriptors and Spectral Properties

The comparative analysis of sound pairs reveals subtle differences in auditory perception, particularly in the relationship between spectral structure and both color and texture qualities. The perception of both texture and color in sound is linked to its spectral structure. Descriptors such as “rough”, “bright”, “dull” and “smooth” consistently correspond to specific spectral characteristics. Spectral stability tends to lead to perceptions of smoothness, while spectral irregularity leads to perceptions of roughness. Similarly, brightness is strongly correlated with high-frequency dominance, while low-frequency components contribute to perceptions of darkness and dullness.

For example, Sounds #7 and #8 have similar color profiles, with darkness values of 25% and 28.6%, respectively, and dullness values of 54.2% and 42.9%. Both sounds also have similar granularity values (25.0% for Sound #7 and 22.7% for Sound #8), but Sound #7 is perceived as significantly rougher (50.0%) than Sound #8 (22.7%). This highlights how variations in waveform regularity and partial distributions across the spectrum affect texture perception. In contrast, Sound #11, which has more stable spectral components, is perceived as smoother (27.8%), while Sound #4, which has an irregular waveform and unstable frequency distributions, is perceived as rougher (50.0%).

Similarly, Sounds #6 and #7, both with structured but non-harmonic spectra, show considerable roughness (22.7% and 50.0%, respectively), suggesting that the dynamics of the spectral structure influence texture perception. In contrast, Sounds #10 and #17, with irregular spectral content, show negligible roughness (5.0% and 6.3%, respectively) but high granularity (50.0% and 56.3%, respectively). This underlines how additional spectral features influence texture perception.

Sound #12, with stable mid-range frequencies, is perceived as brighter (57.9%) and clearer (26.3%), while Sound #14, with a complex high-frequency spectrum, is perceived as both brighter and clearer. This correlation between high frequency components and brightness is consistent across several sounds. In contrast, Sound #11, characterized by a low-frequency profile, is perceived as darker (50%) and duller (16.7%), demonstrating how low-frequency dominance leads to darker color perception. The combined analysis of color and texture descriptors confirms the interdependence of these qualities in shaping auditory perception.

For example, Sounds #5 and #14, both of which have spectral energy concentrated in the higher registers, are perceived as clear. However, Sound #5 is perceived as duller (36.7%), while Sound #14 is perceived as brighter (47.1%). These results suggest that brightness is associated with a structured, near-harmonic spectral organization, while a chaotic spectral organization tends to produce dullness. Similarly, Sound #13, which is characterized by a high degree of striation (33.3%) and a structured spectral distribution, is associated with both clarity and dullness. This is probably due to its mid-low-frequency content and its defined, yet restrained, spectral structure.

Roughness correlates with low-frequency dominance and phase irregularities, while brightness correlates with high-frequency energy. The structure of the sound spectrum—including its distribution, energy, and stability—therefore plays a crucial role in shaping the perception of texture and color.

### 6.2. Metaphorical Use of Cross-Modal Descriptors

This study highlights the role of metaphorical cross-modal descriptors in shaping how listeners articulate abstract sounds. Terms such as “rough” (tactile) and “bright” (visual) illustrate how listeners draw on familiar sensory experiences to describe sound qualities. While these metaphors do not involve literal multisensory perception, they do reveal the cognitive processes involved in auditory comprehension.

Descriptors such as “rough” and “granular” are often used to describe sounds with complex, irregular frequency distributions, evoking tactile impressions. Meanwhile, smooth textures, often associated with continuous and stable spectral structures, evoke specific visual or tactile associations. This finding is consistent with the theoretical framework of embodied cognition, which suggests that metaphorical language emerges from our direct interactions with the physical world, where cognitive processes are rooted in sensory and motor experiences.

The frequent pairing of roughness and granularity reinforces their perceptual and linguistic connection. Rough textures, often associated with low-frequency or irregular sounds, evoke tactile experiences such as rough surfaces, while smoother sounds suggest fluidity and clarity. The consistent pairing of these attributes suggests that listeners use a unified conceptual framework to interpret abstract sounds.

The use of cross-modal descriptors illustrates how sensory experience, linguistic patterns, and cognitive associations shape our perception of abstract sounds. These metaphors not only structure auditory experience, but also highlight the complex interplay between perception, language, and thought.

## 7. Conclusions

This study contributes to the field of auditory perception by investigating how listeners intuitively describe abstract sounds using cross-modal linguistic associations. The results confirm that participants rely on descriptors rooted in other sensory modalities, such as vision and touch, to articulate auditory qualities, reinforcing the idea that sound perception extends beyond strictly auditory parameters.

The results support the notion that listeners use subjective descriptors to articulate the physical properties of sound waves, highlighting the effectiveness of an interdisciplinary framework for sound analysis. The strong correspondence between verbal descriptors and spectral features, particularly brightness, roughness, and granularity, highlights the cognitive mechanisms that shape auditory interpretation in non-referential contexts.

Furthermore, this study demonstrates that abstract sounds—free from identifiable sources or musical structures—provide a unique lens through which to examine auditory perception outside of external referents. This has significant implications for several disciplines, including music analysis, sound design, cognitive psychology, and auditory training. Future research could refine the proposed framework by incorporating neurophysiological data, exploring cultural diversity in auditory descriptions, and extending the methods to real-time interactive sound classification.

This research also contributes to the broader discussion at the intersection of perception, language, and sound analysis. It provides a basis for further studies of how humans structure and interpret auditory experience in different contexts.

## Figures and Tables

**Figure 1 behavsci-15-00396-f001:**
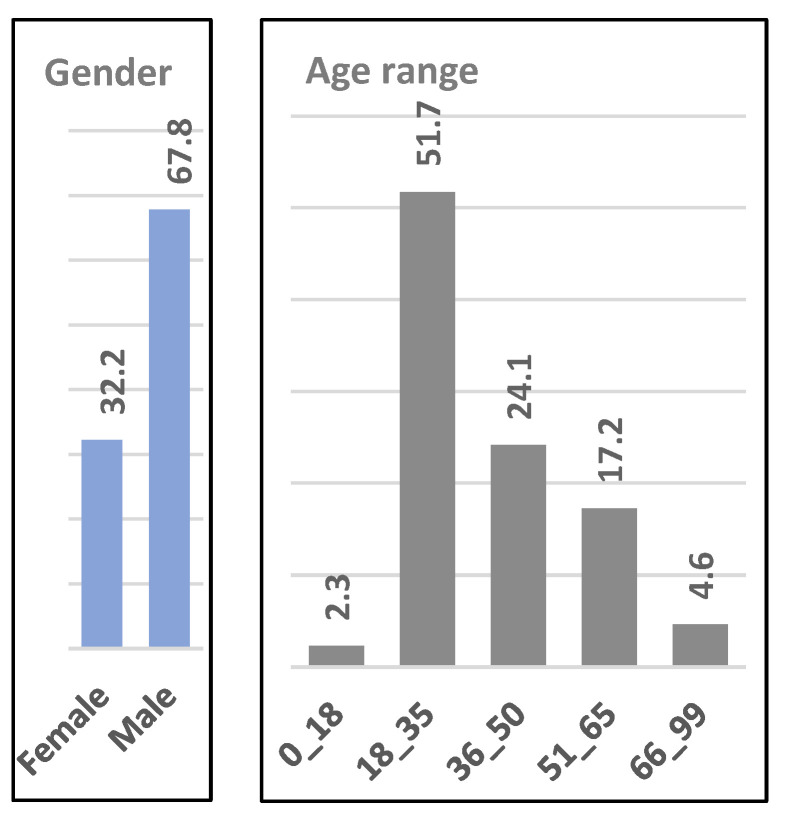
Demographic characterization of participants to the SEQR survey by gender and are range.

**Figure 2 behavsci-15-00396-f002:**
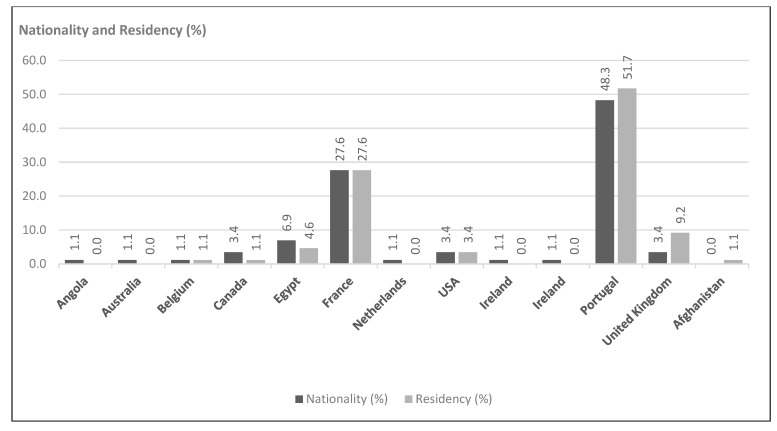
Geographic characterization of participants to the SEQR survey.

**Figure 3 behavsci-15-00396-f003:**
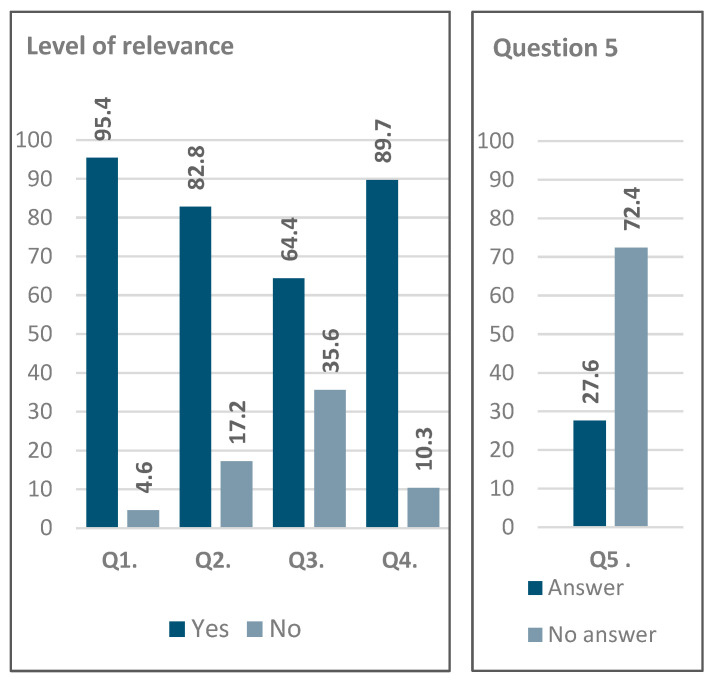
Participants’ answers to SEQR perceptual awareness and application assessment questions.

**Figure 4 behavsci-15-00396-f004:**
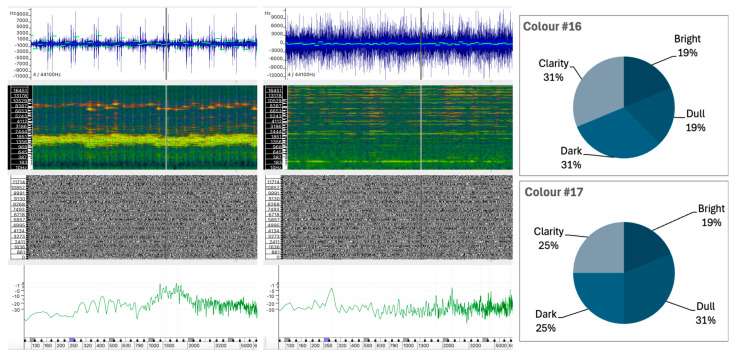
Representation of the physical characteristics for Sound #16 (**left**), Sound #17 (**middle**), and the corresponding perceptual descriptors for color perception results (**right**).

**Figure 5 behavsci-15-00396-f005:**
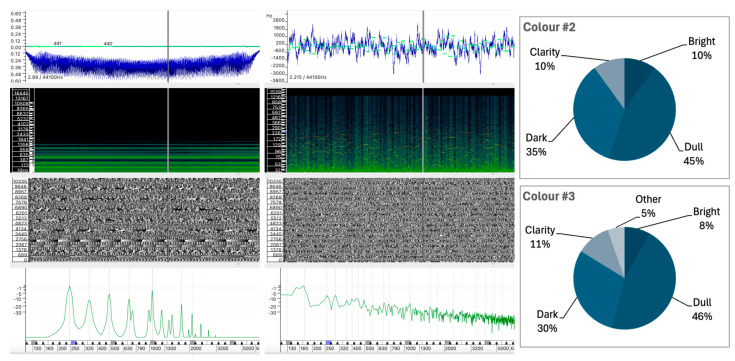
Representation of the physical characteristics for Sound #2 (**left**), Sound #3 (**middle**), and the corresponding perceptual descriptors for color perception results (**right**).

**Figure 6 behavsci-15-00396-f006:**
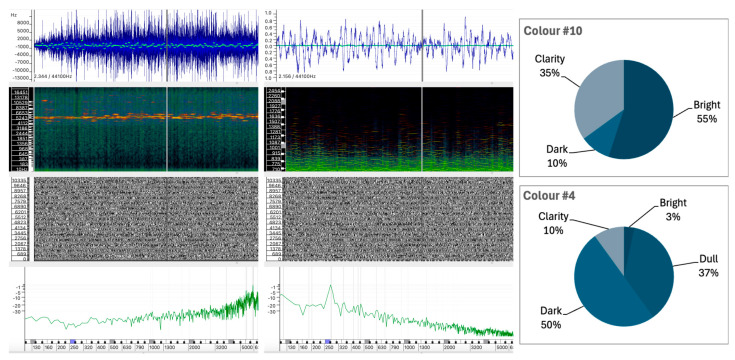
Representation of the physical characteristics of Sound #10 (**left**) and Sound #4 (**middle**), and the corresponding perceptual descriptors for color perception results (**right**).

**Figure 7 behavsci-15-00396-f007:**
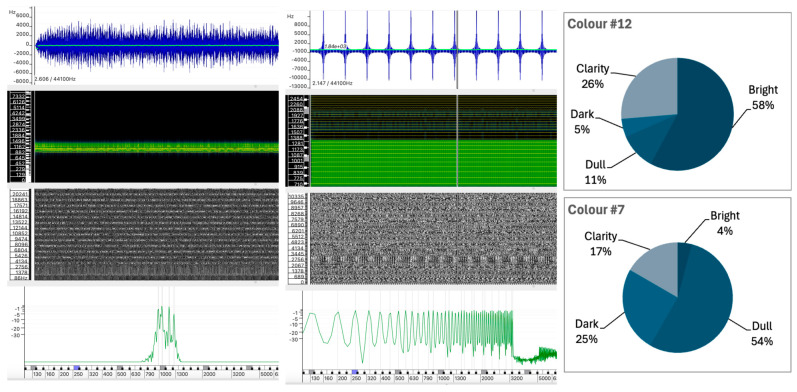
Representation of the physical characteristics of Sound #12 (**left**) and Sound #7 (**middle**), and the corresponding perceptual descriptors for color perception results (**right**).

**Figure 8 behavsci-15-00396-f008:**
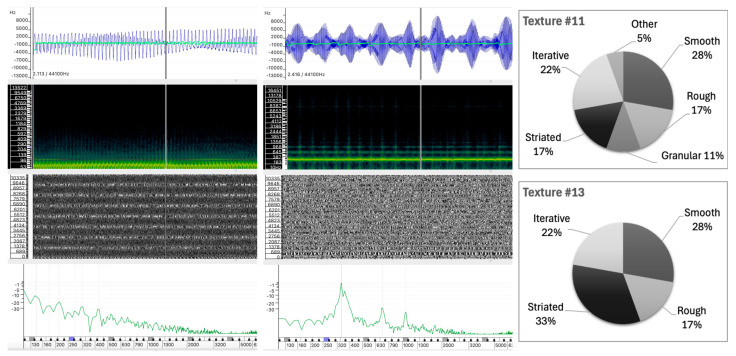
Representation of the physical characteristics of Sound #11 (**left**) and Sound #13 (**middle**), and the corresponding perceptual descriptors for texture perception results (**right**).

**Figure 9 behavsci-15-00396-f009:**
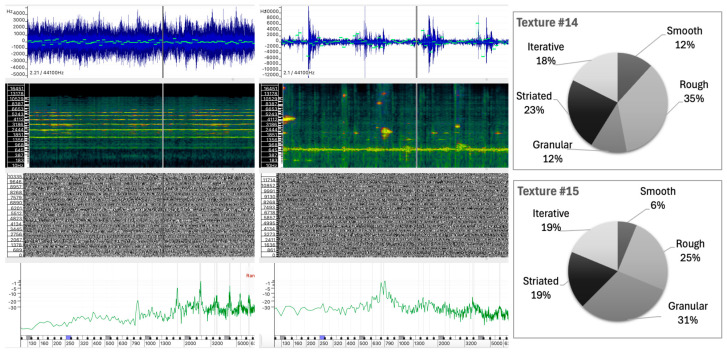
Representation of the physical characteristics of Sound #14 (**left**) and Sound #15 (**middle**), and the corresponding perceptual descriptors for texture perception results (**right**).

**Figure 10 behavsci-15-00396-f010:**
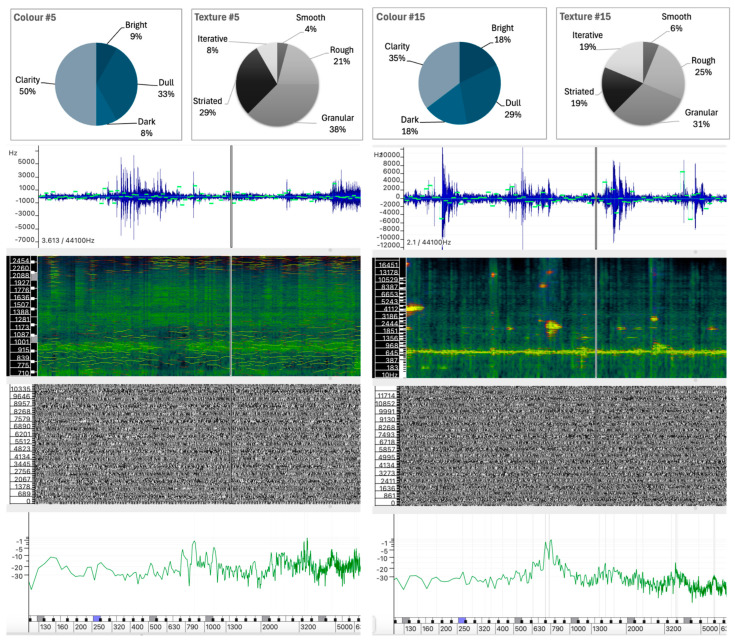
Comparative representation of Sound #5 (**left**) and Sound #15 (**right**) in terms of perceptual attributes and physical characteristics of the sound wave.

**Figure 11 behavsci-15-00396-f011:**
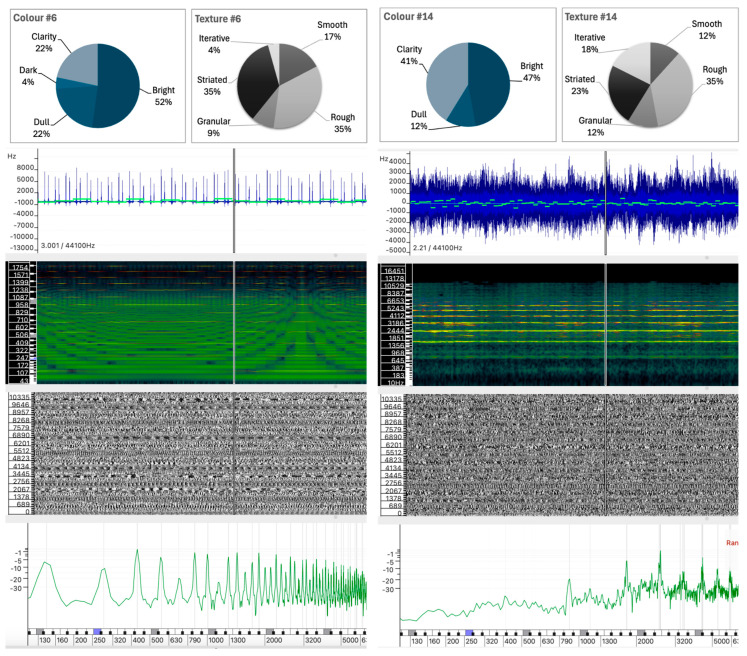
Comparative representation of Sound #6 (**left**) and Sound #14 (**right**) in terms of perceptual attributes and physical characteristics of the sound wave.

**Figure 12 behavsci-15-00396-f012:**
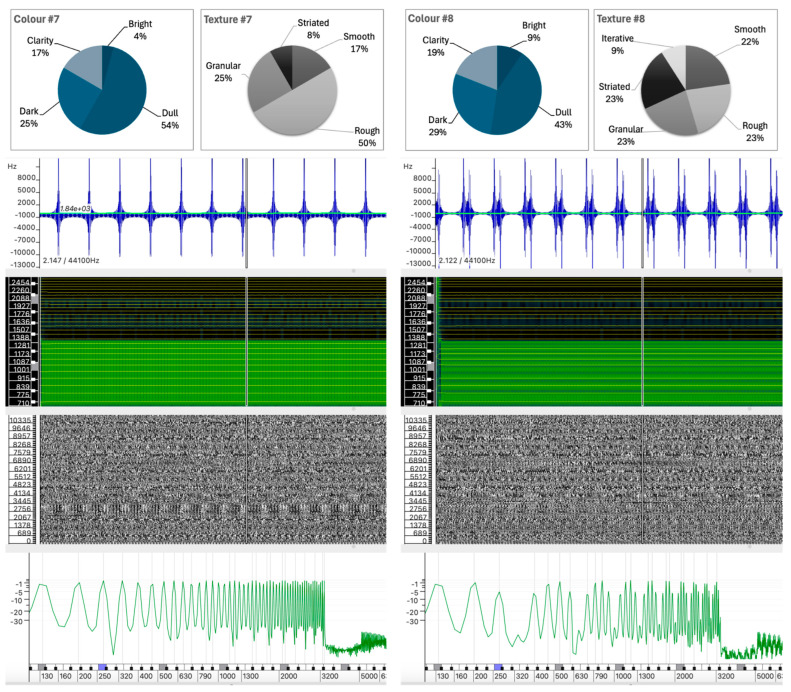
Comparative representation of Sound #7 (**left**) and Sound #8 (**right**) in terms of perceptual attributes and physical characteristics of the sound wave.

**Figure 13 behavsci-15-00396-f013:**
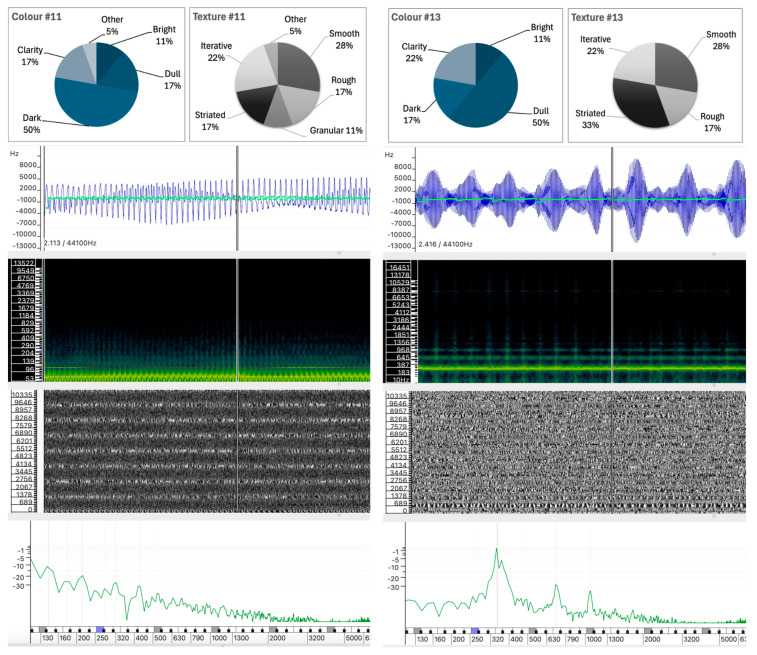
Comparative representation of Sound #11 (**left**) and Sound #13 (**right**) in terms of perceptual attributes and physical characteristics of the sound wave.

## Data Availability

The data are not available publicly; the survey platform does not exist anymore.

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
