# Peer review of "Listening Beyond the Source: Exploring the Descriptive Language of Musical Sounds"

_behavsci, 2025, doi:10.3390/bs15030396_

Round 1
Reviewer 1 Report
Comments and Suggestions for Authors
The paper investigates auditory perception and how people describe abstract sounds with no identifiable source, focusing on the intrinsic qualities of sound waves rather than any associative or referential qualities. Using an interdisciplinary approach, the study highlights a cross-modal basis for auditory descriptors through a semi-large scale, international survey that captured subjective descriptors and their correlation with spectral properties. This research claims to provide a framework for understanding the role of language in describing non-referential auditory stimuli, contributing to potential applications in sound design, auditory training, and music theory.
In general, the idea is clear, although it might be hard to follow throughout the paper. The related work is sufficient and the experimental setup clear.
However, I would prefer to hear the sounds (nevertheless, kudos to the author for describing them individually) and would recommend the author adds a publicly available link for the reader to analyze and possibly re-use the collected sounds.
With regards to the proposed framework, I look forward to the authors future work. I believe it is a bit too early to fully evaluate the impact of the collected data without clearly defined tasks. I would recommend expanding on these in the conclusion section.
Author Response
Thank you for your comments.
Reviewer 2 Report
Comments and Suggestions for Authors
Summary of recommendations
1. Clarify Conceptual Framework: Ensure consistency in discussing cross-modal perception, linguistic conventions, and cultural influences. Define key terms like “abstract sound” with precision.
2. Expand Methodological Detail: Provide comprehensive descriptions of the research tool, sound examples, and selection criteria. Acknowledge the limitations of the study, particularly sampling biases.
3. Enhance Analysis and Visualizations: Use visual tools to present findings, clarify patterns, and ensure all claims are supported by data.
4. Improve Terminology Usage: Avoid misusing terms like “correlation” and “Euclidean distance.” If used, clearly define their application in the study.
5. Extend the Literature Review: Include a more thorough literature review that incorporates perspectives from the semiology of sound and music. This would provide a stronger theoretical foundation and help bridge the gap between linguistic descriptions, cultural context, and perceptual frameworks.
At the beginning of the text and in the abstract, there is a lack of clear information about the author’s intended focus, as it is not explained what “sound” refers to. Is it everything we hear? And what is “abstract sound”? It would be helpful to refer to some classification of sound phenomena (e.g., a classification distinguishing between sounds and noise). In particular, it should clarify the relationship between the studied sounds and music.
Similarly, the article’s perspective is not clearly presented. Initially, sound is discussed in the context of everyday life, but the literature review cites works on the perception of sound in music. Thus, is the focus on a psychoacoustic perspective, the psychology of music, the sociology of music, or the semiology of music?
It is necessary to clarify what “abstract sound” means in Author’s approach, especially regarding the examples used in the research. Additionally, it should be explained how these examples relate to the applied tool in the form of closed-ended questions with predefined characteristics, which are also used to study meanings in music and the reception of sound meanings.
Auditory perception encompasses both neurological processes and the physical properties of sound, as well as the nature of auditory neural responses [see Warren RM. Sound and the auditory system. In: Auditory Perception: An Analysis and Synthesis. Cambridge University Press; 2008:1-34].
Lines 163–165: Perceptual sound qualities are described as abstract potentialities, associated with specific physical characteristics—such as frequency, spectrum, intensity, and spatial position—that shape how listeners differentiate and interpret sounds.
While researchers may classify these as abstract sounds, a critical question arises: to what extent do participants interpret examined sounds without reference to their cultural context? This issue needs further exploration in the analysis.
Line 137: In the context of auditory perception, the justification for the author’s decision to use the concepts of ‘Matter,’ ‘Form,’ and ‘Position’ should be more detailed and explicitly explained. as i understand, this is the main analytical framework.
Line 174: In section 3.2, the author suggests the involvement of other senses in sound perception. However, cross-modal perception is nothing more than the use of specific terms referring to the reception of other senses to describe music. The author herself acknowledges this in other parts of the article. It is important to use terminology more precisely to avoid misleading the reader.
The analysis of verbal descriptors used by listeners reveals that perceptual sound qualities can be organized into three principal fields—matter, form, and position. But why is this the case? Is this conclusion based on the author’s data analysis? It still remains unclear (page 5) what types of sounds are being analyzed and why this framework relates to cross-modal perception. This connection, as promised in the title of section 3.2, is not adequately explained.
In lines 211–220, the author describes theories of sound perception but fails to provide proper citations to indicate who developed these theories and in what research contexts these processes are described. For instance, from a sociological perspective, this process might be described differently, i.e., by considering the influence of cultural context. Factors such as interpretive patterns, associations acquired through socialization with specific sounds, and cognitive schemas shaped by particular cultural settings could be emphasized. Moreover, it still remains unclear what kind of sound is being discussed—music, natural sounds, or something else? This lack of clarity diminishes the precision and applicability of the analysis.
In lines 221–250, the author introduces verbal descriptors, raising the question of how linguistic conventions shaped by specific usages influence the way sound is described. Additionally, it remains unclear why both participants and researchers rely on these particular terms.
The author rather ambiguously addresses the analysis and interpretation of sounds through language that incorporates terms derived from the reception of other senses. She describes this process in a way that suggests we are dealing with the perception of music or sounds through other sensory modalities.
herre are examples:
“line 53 Verbal descriptions of auditory experiences are largely dependent on cross-modal 53 sensory interaction. The findings confirm the usefulness of the intuitive use of descriptors 54 from non-auditory sensory domains to articulate the perception of abstract sounds
line 56 : interaction across sensory modalities in auditory perception
line 82. namely cross-modal interference in auditory perception, intuitive verbal descriptions of sounds (including both abstract and identifiable sounds), and the concept and attributes of perceptive sound qualities.
line 102 that listeners use cross-modal associations, like visual or tactile analogies, to describe auditory experiences
line 118 cross-modal interactions play a pivotal role in shaping auditory perception.
line 134 Sound Qualities in Perception: Cross-Modal Interactions and Fields
line s148-9 Cross-modal sensory fields, linking auditory, visual, and tactile modalities, reveal how 148 listeners intuitively associate sound qualities with other sensory characteristics, shaping auditory experiences. “
However, she repeatedly acknowledges that this is simply about a specific linguistic framework, not the actual use of other senses . After all, sound is perceived through hearing, not touch. This distinction requires greater precision to avoid conceptual confusion. Rather than treating this as an unquestioned fact, the research questions would benefit from a deeper exploration of this phenomenon. Notably, the author references studies examining the terminology experts use to describe music, which could have been utilized to further investigate this issue within the scope of her research. This would provide a more critical and comprehensive analysis.
What is missing is a more in-depth reflection on the development and use of specific terminology to communicate about music, including references to music semiology and linguistic analysis, rather than a reliance on cross-modal references. Are we dealing here with a linguistic and semantic description of sound? why the language used to describe music incorporate vocabulary and associations from other senses? Furthermore, was the earlier discussion not centered on the cross-modal perception of music? This apparent shift in focus requires clarification and consistency to strengthen the overall argument.
Visualisations needs improvements:
fig. 1 Demographic characterisation of participants . The percentages for gender do not sum up to 100%. Why is this the case? Is it due to missing data, multiple responses, or a categorization issue? This discrepancy should be clarified in the figure or accompanying tex
fig 3 the categories are misleading,: the labels “Yes,” “No,” “Answer,” and “No answer” represent two different types of categories: binary response categories. and response rate; mixing these categories in a single chart creates confusion; The same axis is used to represent “Yes,” “No,” “Answer,” and “No answer,” yet these terms seem to refer to different dimensions (e.g., response type vs. participation).
Misleading use of colors: “Yes” and “No” are represented with the same shade of blue, which creates ambiguity.
Recommendations :
- it is better to use contrast colours (black -withe, blue- red )
-Split “Yes/No” and “Answer/No answer” into separate charts
Methodology
The description of the methodology is insufficient.
Overall, there are substantial concerns regarding the methodological description, including the presentation of the research tool, clarity of the research procedures, and the analytical framework. The methodology section should be expanded to include a more detailed description of the research procedure and explicit acknowledgment of the study’s limitations; particularly regarding the sounds and its descriptions from a predefined list - should be presented with detailed explanations to ensure clarity about the research tool used.
A critical question arises: why are variables such as gender, age, and nationality presented in the graphs when these factors are not utilized in the subsequent analysis? The data analysis primarily focuses on the frequency of indicating specific sounds’ features, making the inclusion of demographic variables in visualizations appear unnecessary.
Additionally, the study sample is small (n=70), non-random, and self-selected, leading to significant biases. The limitations of such a sampling method are well-known and should be acknowledged.
Analysis
The data analysis provided is too brief, and the results lack adequate visualization. Graphical representations or other forms of visualization are necessary to illustrate the patterns and trends discovered in the data; only selected result are visualised.
Several questions remain unanswered fully: What accounts for the similarities or differences in the participants’ selections of answers for a given sound? A clearer explanation is needed to reveal the rationale behind these outcomes.
Some conclusions are debatable, considering the previous remarks:
Line 737: “Descriptors such as ‘rough’ and ‘bright’ consistently aligned with specific spectral characteristics, suggesting that listeners intuitively interpret auditory qualities through multisensory experiences.”
It remains unclear whether these “multisensory experiences” are truly factors shaping the perception of sound, or if they merely reflect the language available to participants for describing sound. Referring to semiology could provide valuable insights and help clarify whether the observed patterns arise from perceptual mechanisms or linguistic conventions. Additionally, referencing studies on the universality of meaning units in music perception could shed light on whether these descriptors and their associations have broader cultural or cognitive relevance.
Key methodological issues and recommendations
1.Insufficient explanation of the methodology
•The logic behind the selection of examples for detailed analysis is unclear. For instance, only half of the examples are analyzed for attributes like color and texture. Why were not all examples analyzed for each attribute?
•There is no explanation of how paired examples were selected for comparison (e.g., examples 16 & 1, 2 vs. 3, 11 vs. 13). The criteria for selection are not provided, and from the table, it is evident that the pairs do not consistently represent similarity or dissimilarity. What was the guiding logic behind these choices?
2.Mode of Presentation
The descriptive analysis used in the text is overly dense and makes it challenging for readers to follow the analysis. The inclusion of more visualizations, such as graphs or diagrams, would significantly enhance clarity and help illustrate the perception patterns that the author identifies.
By addressing these issues, the analysis would be more comprehensible and provide clear insights into the patterns of the data.
3. Analysis.: what actually is ment by correlation and Euclidean distance?
The author primarily relies on differences in the percentage of indications for specific features and the differences in percentage-based Author calls: “Euclidean distance.” Usually, ‘Euclidean distance’ is used as a metric in multidimensional spaces ; in this case, it is indeed one-dimenision distance; In some cases, authors use “Euclidean distance” as a loose term to mean “distance” or rather “difference,” which can lead to potential misinterpretations.
The author describes sound perceptions using more than one feature, strictly speaking, and the author could apply a multidimensional formula for Euclidean distanc,
Furthermore, “correlation” is mentioned several times but no actual correlation analysis (e.g., Pearson or Spearman coefficients) was performed, the term is being used loosely, potentially misleading readers. Instead, the comparisons are based solely on percentages.
4.Given the small sample size and the large number of categories, it is unclear whether the observed differences genuinely indicate distinct types of perceptual reception or if they are merely coincidental. This ambiguity calls for a more robust statistical approach to validate the findings and distinguish between perceptual similarities and random variations.
5.The charts representing the physical characteristics of sound are insufficiently explained. Most lack a legend, making it difficult to interpret the data. Additionally, the physical characteristics of the sound waves are only roughly described in the text. These characteristics should be clearly defined, including a detailed explanation of what they represent, how they are interpreted, and, most importantly, how the measurements were produced.
6. Furthermore, Table 5 includes only a selection of examples for analysis. The criteria for selecting these specific examples over others are not provided. This omission raises questions about the rationale behind the selection process and whether it was systematic or arbitrary. A clear explanation of the selection criteria is essential to ensure transparency and the validity of the analysis.
This fragment is not clear: line 304-5: Amateurs: Non-professional musicians could specify genres (e.g., classical, electroacoustic, pop, jazz). Professionals: Music or sound professionals could indicate roles (composer, per-306 former) and primary musical genres.
Lines 295 and 301: Avoid repetition—combine the two sentences for clarity:
The data include demographic details such as age, gender, nationality, and residence, as well as participants’ education level, field of study, and professional background.
Line 412: Correct the reference to the table:
Table XXX should be updated to Table 2.
Line 506: Correct the reference to the table:
Table YYY should be updated to Table 3.
Author Response
Comment 1: Thank you for your thoughtful feedback regarding the conceptual framework and the definitions of key terms such as "abstract sound" and "cross-modal perception." I recognize that additional precision and explicit definitions will improve the text’s clarity and ensure readers fully understand the study's focus. n particular, the term “abstract sound” is used in this study to refer to synthesised auditory stimuli intentionally designed to exclude causal indices and source references, directing the listener’s attention toward intrinsic sound qualities rather than associating the sounds with musical or environmental contexts. These sounds are not musical compositions but can be considered as materials with potential musical applications.
Comment 2: Thank you for highlighting the need for more comprehensive methodological details, have expanded the methodology section to include detailed descriptions of the research tool, the selection process for the sound examples, and their characteristics. Additionally, I have explicitly acknowledged the limitations of the study, particularly the self-selected nature of the sample, which may introduce biases. The term "cross-modal perception" in the context of this article refers to the intuitive use of descriptors from non-auditory sensory domains (e.g., tactile or visual) to articulate auditory experiences. While these terms were introduced and applied, I agree that further elaboration will strengthen the manuscript. To this end, I have revised the introduction and methodology sections to explicitly define these terms and clarify the relationship between the studied sounds and their broader context.
Comment 3: I improved the visualizations and clarify what means each layer of the images, as well as the techniques uses to sound analysis.
Comment 4: I clarify the use of Euclidean distance and include a table showing the euclidean distance between pairs of sounds, combining the colour ans texture perceptive results.
Comment 5: Thank you for your observation regarding the inclusion of auditory physiology and cognitive processes. While these aspects are indeed integral to auditory perception, this study specifically focuses on the verbal descriptors used to articulate auditory experiences. This decision reflects the study’s aim to contribute to the understanding of sound semantics rather than the underlying biological or cognitive mechanisms. I have clarified this distinction in the manuscript to ensure the scope is clear to readers.
Thank you for your suggestion to include aspects related to the neurophisiological field of the auditory perception, but in fact, my goal with this research is not to make a direct relationship between the auditory system and the neurophisiological responses, since it would be another full article, but instead I want to be focused on People naturally articulate and describe sounds using words in their daily lives, relying on intuitive and spontaneous expressions that do not require the structured conditions of a controlled laboratory environment.
I include also a note explaining the choice on the theoretical direction given to the article, (including my intention to not enter on the field of semiology ).
Nevertheless, I effectively extended the literature review
Thank you for your observation regarding the inclusion of auditory physiology and cognitive processes. While these aspects are indeed integral to auditory perception, this study specifically focuses on the verbal descriptors used to articulate auditory experiences. This decision reflects the study’s aim to contribute to the understanding of sound semantics rather than the underlying biological or cognitive mechanisms. I have clarified this distinction in the manuscript to ensure the scope is clear to readers.
Thank you for your comment on perceptual sound qualities associated with specific physical characteristics, and the influence of culture on the use of descriptors.
Thank you for your observation regarding the potential influence of cultural context on participants' interpretation of auditory sensations. While it is natural for listeners to interpret auditory experiences through the lens of their own cultural and experiential background, this study does not aim to analyze these cultural influences explicitly. Instead, it focuses on the perceptual descriptors themselves and the associations participants made with specific physical sound characteristics.
That said, I acknowledge that most participants in this study are from predominantly Western cultural contexts, which may reflect a relatively stable cultural framework. I have added a note in the discussion to acknowledge this limitation and suggest that future research could explore cultural variability in sound description.
Thank you for pointing out the need to clarify why only 70 surveys were validated. In the manuscript, I mentioned this, but I realize that further explanation would help ensure transparency. To address this, I have expanded the methodology section to explicitly describe the validation process and the rationale for including only 70 surveys in the analysis.
I should also say that I appreciated very much the detailed review of all flaws and typos you found in the manuscript. I corrected them with care. Just to not that in the first figure, is was effectively an error on the calculation formula: it makes no sens that a percentage counts on a total of 50.